**Cite this article:** von Hoermann C, Weithmann S, Deißler M, Ayasse M, Steiger S. 2020 Forest habitat parameters influence abundance and diversity of cadaver-visiting dung beetles in Central Europe. *R. Soc. open sci.* **7**: 191722.

ecology/environmental science

carrion decomposition, ecosystem service, forest understorey, land use, geotrupidae, vascular plant diversity

**Author for correspondence:**
Christian von Hoermann
e-mail: christian.vonhoermann@gmail.com

# Forest habitat parameters influence abundance and diversity of cadaver-visiting dung beetles in Central Europe

Christian von Hoermann[1,2], Sandra Weithmann[3], Markus Deißler[3], Manfred Ayasse[3] and Sandra Steiger[4]

[1]Department of Wildlife Ecology and Management, University of Freiburg, Tennenbacher Str. 4, 79106 Freiburg, Germany
[2]Department of Visitor Management and National Park Monitoring, Bavarian Forest National Park, Freyunger Str. 2, 94481 Grafenau, Germany
[3]Institute of Evolutionary Ecology and Conservation Genomics, Ulm University, Albert-Einstein Allee 11, 89069 Ulm, Germany
[4]Department of Evolutionary Animal Ecology, University of Bayreuth, Universitätsstraße 30, 95447 Bayreuth, Germany

CvH, 0000-0001-6487-1540

Dung beetles provide crucial ecosystem services and serve as model organisms for various behavioural, ecological and evolutionary studies. However, dung beetles have received little attention as consumers of large cadavers. In this study, we trapped copronecrophagous dung beetles on above-ground exposed piglet cadavers in 61 forest plots distributed over three geographically distinct regions in Germany, Central Europe. We examined the effects of land use intensity, forest stand, soil characteristics, vascular plant diversity and climatic conditions on dung beetle abundance, species richness and diversity. In all three regions, dung beetles, represented mainly by the geotrupid species *Anoplotrupes stercorosus* and *Trypocopris vernalis*, were attracted to the cadavers. High beetle abundance was associated with higher mean ambient temperature. Furthermore, *A. stercorosus* and *T. vernalis* were more abundant in areas where soil contained higher proportions of fine sand. Additionally, an increased proportion of forest understorey vegetation and vascular plant diversity positively affected the species richness and diversity of dung beetles. Thus, even in warm dry monocultured forest stands exploited for timber, we found thriving dung beetle populations when a diverse understorey was present. Therefore, forestry practices that preserve the understorey can sustain stable dung beetle populations and ensure their important contribution to nutrient cycles.

# 1. Introduction

Detritus decomposition is fundamental to the functioning and preservation of ecosystems. Plant-derived non-living organic matter and animal cadavers comprise the basal trophic level of many food webs. Thus, detritus has a huge impact on ecosystem properties such as nutrient cycles and biodiversity [1–4]. A wealth of information is available on the complex ecological linkages of plant detritus, whereas the implications of degrading animal biomass are far less understood [5,6]. The process of carrion decomposition returns a considerable amount of nutrients into the surrounding environment and affects a wide range of microbes, vertebrates and invertebrates at several trophic levels [7]. Among the various consumers of cadaver resources, insects, in particular, provide crucial ecosystem services by recycling and dispersing cadaver nutrients [8]. Consequently, the identification of factors influencing carrion-associated insect communities is of great importance in the research field of carrion ecology.

In general, cadaver-associated insects that colonize and break down the cadaver into its respective nutrients do so in a predictable pattern of succession [9]. Calliphoridae and Sarcophagidae flies are the first insects to arrive at a fresh cadaver and their offspring are the main insect inhabitants when the cadaver starts to bloat [10]. During the so-called post-bloating stage of cadaver decomposition, the large feeding masses of fly maggots are joined by predatory coleopterans, i.e. by some members of the Silphidae, Staphylinidae and Histeridae. In the stages of 'advanced decay' and the 'dry remains stage', other coleopterans of the Cleridae and Dermestidae dominate the fauna of the cadaver [10,11]. Apart from these typically anticipated carrion insects, additional coleopteran species of the Scarabaeoidea (Aphodiidae, Scarabaeidae and Geotrupidae) are attracted to a cadaver in its final stages of decay.

The dung-associated visitors are rather peculiar occupants of a carrion resource, as the majority of dung beetles use mammalian dung as a resource for food and breeding [12]. Thus, although copronecrophagous or necrophagous dung beetle species are known to exist and colonize cadavers [13], they have received little attention in carrion research to date. Only recently have studies started to examine the impact of dung beetles on carrion resources. For instance, high numbers of the typical forest dung beetles *Anoplotrupes stercorosus* (Scriba, 1791) and *Trypocopris vernalis* (Linnaeus, 1758) (both Geotrupidae) were observed to be attracted to medium-sized pig cadavers in Central European forests [14]. Similarly, by far the most abundant beetle species found on five exposed deer carcasses in the Sonian Forest in Belgium was *Geotrupes stercorarius* (Geotrupidae) [15]. Hence, dung beetles might contribute significantly to carrion decomposition and nutrient cycling. Vice versa, cadavers might be an important resource required to sustain large dung beetle diversity.

The taxon Geotrupidae (also referred to as earth-boring dung beetles) consists of more than 600 species that are distributed worldwide, mainly in drier areas [14]. In Central Europe, only a few members of the taxon Geotrupidae with large body sizes between 20 and 30 mm are represented by the taxon Geotrupinae. *Anoplotrupes stercorosus* and *T. vernalis* are typical forest geotrupids with common occurrences in lowlands and mountains of Europe [14]. *Anoplotrupes stercorosus* reaches western Siberia and *T. vernalis* was noted in Asia Minor [16]. In the taxon Geotrupinae, diverse feeding behaviours exist like saprophagy, mycetophagy, phytophagy and coprophagy [12,17]. *Anoplotrupes stercorosus*, as a very frequent forest species, can be found in all kinds of dung, on carrion and in old fungi [18]. *Trypocopris vernalis* is common throughout sandy areas (including the East Frisian Islands in Germany) and can be found mainly on the dung of horses and rabbits [18]. Geotrupinae adults excavate nests for larval development under or near the food resource and provide parental care, including food provisioning and feeding of their larvae [12]. Regarding habitat preferences, Jarmusz and Bajerlein [14] showed high numbers of *A. stercorosus* and *T. vernalis* on pig cadavers in hornbeam-oak forests and their lowest abundance in alder forests. In general, *T. vernalis* as a light-seeking species is more frequent in clear-cut areas, forest plantations and brownfields on the edge of forests compared with old-growth and mature stands [19]. However, on pig cadavers used as bait, Jarmusz and Bajerlein [14] showed a high abundance of *T. vernalis* in pine-oak forests and hornbeam-oak forests, what is contrary to previous observations in this species.

The lack of investigation into the role of dung beetles in carrion ecology is all the more surprising as these beetles are the focus of a wide range of behavioural, ecological and evolutionary studies [20–22]. As a globally distributed insect group, dung beetles provide several key ecosystem functions, such as secondary seed dispersal, nutrient cycling and parasite suppression [23]. Additionally, dung beetles are particularly sensitive to anthropogenic disturbances of natural habitats and can be used as biological indicators in monitoring programmes [24–27]. Especially in the tropics, forest dung beetle species are usually unable to tolerate open environments and do not persist after the native forest is replaced by crops and/or cattle systems [28,29]. In Central Europe, the activities of dung beetles are affected by

anthropogenic habitat conversion from forests to grasslands and by land use intensification within forests and grasslands. In forests, for instance, the amount of harvested timber reduces dung removal rates by 20% [27].

For cadaver-inhabiting dung beetles, a large gap of knowledge currently exists regarding the effect of land use and other habitat parameters on diversity, species richness and abundance. Barragán *et al.* [30] were the first to find evidence that copronecrophagous dung beetle species are sensitive to anthropogenic effects. Using traps baited with rotting fish, they observed a decline in the functional diversity of copronecrophagous dung beetle communities as a result of changes in human land use [30].

To increase our knowledge of dung beetles in terms of carrion ecology, we conducted a large-scale study in the framework of the German Biodiversity Exploratories whereby we exposed 75 piglet cadavers across differently managed forest stands in Central Europe (see also [31]) and monitored cadaver-visiting dung beetles during the whole course of decomposition. We expected to trap higher numbers of dung beetles at carrion exposition sites compared with control sites. In this regard, we also anticipated variations in the abundance of trapped beetles in association with specific decomposition stages. As we trapped beetles in diverse forest habitats, we furthermore predicted to find differences in beetle abundance and diversity with differing abiotic and biotic environmental parameters, such as location, forest structure (main tree species, stand age and density, crown closure, composition and coverage of understorey vegetation), soil parameters (type, temperature, moisture and density), climatic conditions (humidity and temperature) and land use intensity. Owing to physiological restrictions and living habits, dung beetles are highly sensitive to changes in temperature and humidity [32–35], and soil properties [14,36]. Thus, we particularly expected to observe variations in the occurrence of dung beetles with differing climatic and soil conditions.

# 2. Methods and materials

## 2.1. Study regions

Within the conceptual framework of the Biodiversity Exploratories, we performed the present study in three distinct regions in Germany, varying significantly in geography (for more conceptual details, see http://www.biodiversity-exploratories.de). The three regions are Biosphere Reserve Schwäbische Alb (German state of Baden-Württemberg) in the southwest (48°20′60.0″ N to 48°32′3.7″ N; 9°12′13.0″ E to 9°34′48.9″ E), Hainich-Dün (German state of Thuringia) in the middle (50°56′14.5″ N to 51°22′43.4″ N; 10°10′24.0″ E to 10°46′45.0″ E) and Schorfheide-Chorin (German State of Brandenburg) in the northeast (52°47′24.8″ N to 53°13′26.0″ N; 13°23′27″ E to 14°8′52.7″ E). A more precise description of the study areas can be found in von Hoermann *et al.* [31].

## 2.2. Study sites and piglet cadaver exposition

Seventy-five experimental plots within forests (EPs, 100 × 100 m, 25 per region) were chosen based on a stratified random design. These strata support land use and some other factors like type and depth of soil [37]. All plots represented a similar range of silvicultural application intensities of the characteristic soil types in each region [37]. For measuring land use and intensity, forest management intensity was predicted by using silvicultural management intensity (SMI). This precalculated indicator combines the following three characteristics: tree species, stand age (risk of stand loss component) and over ground, dead and vital woody biomass (stand density component) [38]. The risk component defines the combined effect of tree species selection and stand age on SMI and the stand density component quantifies the effect of removals and regeneration method using actual biomass related to a reference [38]. Schall and Ammer [38] stated that SMI at the operational level is mostly related to fellings (harvest operations, thinning and tending), but in the case of trees remaining in the stand due to natural losses such as windthrow, the discrepancy between removals and fellings becomes more clear. They commented that removals (used for SMI description in the risk of stand loss component) are more indicative of forestry intensity than trees that are lost due to natural or silvicultural reasons [38]. Schall and Ammer [38], therefore, suggested to measure removals by the deviance between maximum biomass (age, site and species-specific) and actual biomass of dead and living trees.

From 4 August till 4 September 2014, we laid out 75 cadavers of stillborn piglets (species: *Sus scrofa domestica* with an average weight of 1.44 kg) on 25 EPs per region (one piglet per forest plot). EPs were sufficiently spaced to account for uninfluenced carcasses. We applied piglet cadavers because their insect

visits and general arthropod succession are very well studied [39–42]. Additionally, their wild-type (wild boar; *Sus scrofa*) is present Germany-wide in numerous forest habitats. Since the study was focused on carrion insects, all piglets were sheltered in wire cages (63 × 48 × 54 cm; MH Handel GmbH, Munich, Germany) to exclude scavenging by larger mammalians or avian species. Wire cages containing carcasses and controls (pitfall traps without cadavers and wire cages) were installed at a distance of 100 m from each other. A more detailed description of the exposure of the piglet cadavers is given in von Hoermann *et al.* [31].

## 2.3. Trapping of cadaver insects

At each exposed carcass, two pitfall traps (one at the head and one at the anus) were buried for capturing cadaver insects. Head and anus are the two most important infestation sites for carrion insects [43]. Pitfall traps (diameter: 95 mm) were opened for 48 h to ensure a constant sample period for each trapping event. To reduce surface tension, we used an odourless soapy solution (one drop of detergent, Klar EcoSensitive, AlmaWin, Winterbach, Germany). A more detailed description of the pitfall trap installation and beetle sampling is given in von Hoermann *et al.* [31]. In the present study, we focused on dung beetles (mainly Coleoptera: Geotrupidae) attracted to our piglet cadavers. All dung beetle individuals were identified to the species level [44–46]. To identify specimens of Aphodiidae and Scarabaeidae taxa, we examined male genital tracts. After species determination, we stored all individuals at Ulm University. Despite losses of piglets on four plots and restrained entry for particular sampling days on 10 plots in highly conserved areas, the overall sample collection revealed 854 samples from 61 forest plots as a data basis for statistical analyses [47].

## 2.4. Environmental parameters

Twenty-one biotic and abiotic environmental parameters were used for statistical analyses in order to determine important factors that affect the abundance of dung beetles attracted to carrion. Twenty variables and their respective values were obtained from the BExIS platform (Biodiversity Exploratories Information System, https://www.bexis.uni-jena.de) for each EP (electronic supplementary material, table S1): 'location', 'management system' [48], 'main tree species' (MTS) [49], 'air humidity' [50], 'stand age' [51], 'stand density MTS' [52], 'crown closure' [49], 'forest understorey' [53], 'dbh standard deviation' (standard deviation of diameter at breast height (dbh)) [52], 'Shannon total vascular plants' (Shannon diversity index for all vascular plants) [54], 'Simpson total vascular plants' (Simpson diversity index for all vascular plants) [54], 'soil type' [49], 'soil temperature' [50], 'soil moisture' (soil moisture at 10 cm below surface) [50], 'mineral soil pH' [55], 'bulk density' [56], 'fine sand' (particle size: 0.063– 0.2 mm) [57], 'fine silt' (0.002–0.0063 mm) [57], 'clay' (less than 0.002 mm) [57] and 'forest utilization intensity SMI' [38,58]. Variables obtained from the BExIS platform were collected across each 100 × 100 m plot. Their values were known from multiple inventory campaigns performed within the Biodiversity Exploratories (basic data including soil composition and type, bulk density, vertical structure, management and climate). Additionally, we included the 'mean ambient temperature' measured with data loggers (Thermochron iButton, Whitewater, WI, USA) in the immediate vicinity of each cadaver in 30 min intervals during the entire time of fieldwork [47].

## 2.5. Statistics

All models were calculated in R v. 3.3.1 [59]. By univariate approaches (quasi-Poisson-GLM, Gaussian-GLM), we investigated differences in species diversity (Simpson's dominance and Shannon's diversity for the Schorfheide-Chorin exploratory), richness and total abundance of dung beetles among the various forest habitats. To get an exhaustive knowledge of the mechanism that shifts in rare and abundant species push interactions [60], we calculated Shannon's diversity (sensitive to rare and abundant species to the same extent [61]) and Simpson's dominance index (responsive to abundant species, more general compared with Simpson's diversity [62]), in addition to species richness (sensitive to rare species [60]) for the scarabaeoid beetle taxon. Differences in species diversity, richness and total abundance were calculated for the whole scarabaeoid beetle taxon and also separately for total abundance of the two most abundant single geotrupid species *A. stercorosus* and *T. vernalis* (both geotrupid taxa represent 99.86% of all collected specimens compared with only 0.14% represented by the taxa Aphodiidae and Scarabaeidae; figure 1*a*). Because *T. vernalis* was only present in the Schorfheide-Chorin region, we conducted all R-analyses for this species solely based on the data obtained from the 24 experimental plots of the

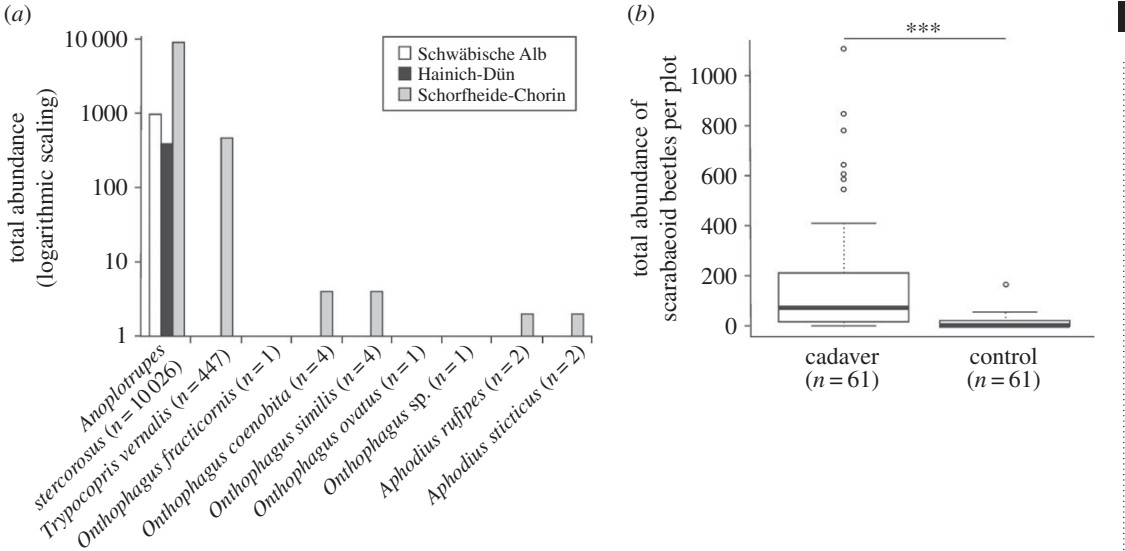

**Figure 1.** (*a*) Total abundance (logarithmic scaling, base 10) of individuals of trapped scarabaeoid dung beetle species (separate for each exploratory) during the exposure of 61 piglet cadavers from 4 August till 4 September 2014 (*n* represents the total number of beetles caught during the course of this study). (*b*) Abundance distribution of trapped scarabaeoid dung beetles per plot in all three exploratories. A significant difference of trapped scarabaeoid dung beetles is found between control and piglet cadaver traps. Each box shows the median, the 75% percentile, the 25% percentile, the highest non-extreme value, the smallest non-extreme value and the extreme values inside a category (Kruskal–Wallis test: $\chi^2 = 37.97$, d.f. = 1, ***$p < 0.001$, *n* represents the number of sampling plots).

Schorfheide-Chorin exploratory. Similarly, the two diversity indices were calculated and evaluated exclusively for 24 data rows of the Schorfheide-Chorin region. This was because of the presence of only one single species (*A. stercorosus*) in pitfall traps, surrounding piglet cadavers, on forest plots of the Hainich-Dün and Schwäbische Alb exploratories. In the cases of overdispersion, quasi-Poisson error distributions were calculated in the corresponding models [63]. To test the effects of trap type (piglet cadaver traps versus control traps) and decomposition stage on overall abundance of dung beetles, Kruskal–Wallis rank sum tests (including Tukey tests for *post hoc* pairwise comparisons) were applied.

For predicting the relative importance of environmental parameters (electronic supplementary material, table S1) on species richness, diversity and on total dung beetle abundance, we calculated a random forest (randomForest function in the package MASS) for identification of the seven most important environmental variables out of all variables considered in this study (electronic supplementary material, tables S2, S4 and S6) (following [64,65]). Forest utilization intensity (SMI) was taken into account in separate quasi-Poisson- or Gaussian-GLMs (the latter are valid for diversity indices as non-integer dependent variables) in order to avoid linear dependency attributable to the combination of the following parameters: main tree species, stand age and over ground, dead and vital woody biomass [38]. The above-mentioned random forest algorithm works as a recursive partitioning and classification tree method [66]. It is based on a so-called forest of regression trees and it uses random inputs [67,68]. *A priori*, we fitted the seven most important variables (derived from the random forest output) in quasi-Poisson- or Gaussian-GLMs in a sequence in line with their relevance (electronic supplementary material, tables S3, S5 and S7 (after [65])). For quasi-Poisson-GLMs, subsequent stepwise model simplification (stepAIC function in the MASS package) in both directions (backwards and forwards, starting with the full model) was performed (after [65]), based on the quasi-Akaike information criterion (QAIC). For Gaussian-GLMs, the same procedure was performed based on the Akaike information criterion (AIC).

# 3. Results

## 3.1. Beetle abundance

Throughout the field trial, we captured 10 488 individuals from the following nine scarabaeoid beetle species in pitfall traps surrounding 61 piglet carcasses: *Anoplotrupes stercorosus* (*n* = 10 026), *Trypocopris vernalis* (*n* = 447), *Onthophagus coenobita* (*n* = 4), *Onthophagus similis* (*n* = 4), *Aphodius rufipes* (*n* = 2),

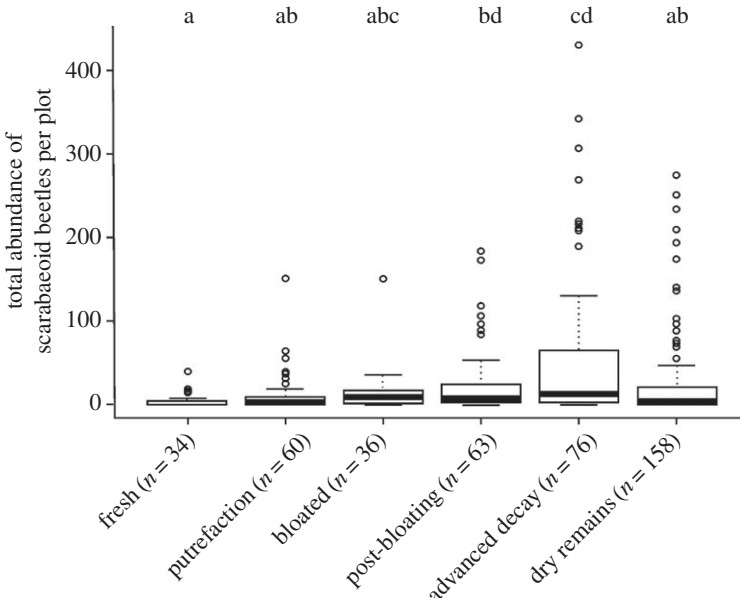

**Figure 2.** Median abundance of trapped scarabaeoid dung beetles per plot for each decomposition stage across all three exploratories. The different letters indicate significant differences between decomposition stages (Kruskal–Wallis test: $\chi^2 = 30.04$, d.f. = 5, $p < 0.001$; Tukey tests ($p < 0.05$; ~ indicates 'against'): post-bloating ~ fresh, $p = 0.006$; advanced decay ~ fresh, $p < 0.001$; advanced decay ~ putrefaction, $p = 0.005$; dry remains ~ advanced decay, $p = 0.007$; $n$ = number of sampling events).

*Aphodius sticticus* ($n = 2$), *Onthophagus fracticornis* ($n = 1$), *Onthophagus ovatus* ($n = 1$) and *Onthophagus* sp. ($n = 1$; figure 1$a$). In the respective controls, we only trapped a total of 863 individuals of the species *A. stercorosus* ($n = 827$) and *T. vernalis* ($n = 36$). We monitored a large variation in numbers of individuals and species on cadavers between the plots. The plot range was from 0 (two single plots in Hainich-Dün) to 1109 (one particular plot in Schorfheide-Chorin) individuals. The number of species per plot varied from zero captured dung beetle species on two sampling plots in Hainich-Dün to five trapped species on one plot in Schorfheide-Chorin.

## 3.2. Effects of decay stage and trap treatment on dung beetle abundance

Across all three exploratory regions, cadaver-associated traps captured considerably more dung beetles by contrast with unbaited control pitfall traps (Kruskal–Wallis test: $\chi^2 = 37.97$, d.f. = 1, ***$p < 0.001$; figure 1$b$). Furthermore, cadavers in advanced decay attracted, across all three regions, significantly more dung beetle individuals than those that were fresh, putrefied or dried-out (Kruskal–Wallis test: $\chi^2 = 30.04$, d.f. = 5, $p < 0.001$; Tukey tests: $p < 0.05$; figure 2).

## 3.3. Influence of environmental variables on overall dung beetle abundance

From a total of six predictors in the reduced model (quasi-Poisson-GLM: $F = 16.84$, $p < 0.001$), 'mean ambient temperature' considerably influenced the abundance of dung beetles, showing an expansion of total dung beetle abundance under the condition of higher temperature values in all three regions (figure 3; electronic supplementary material, table S3$a$). The simplified model explained 70.6% of the variance. Forest management represented by the forest utilization intensity index (SMI) had no effect on overall dung beetle abundance (quasi-Poisson-GLM: $F = 0.41$, $p = 0.524$).

## 3.4. Influence of environmental variables on the abundance of the single geotrupid species *A. stercorosus*

From a total of six predictor variables in the simplified model (quasi-Poisson-GLM: $F = 15.30$, $p < 0.001$), higher values of the abiotic parameters 'mean ambient temperature' and 'fine sand' considerably raised the abundance of *A. stercorosus* individuals in all three regions (electronic supplementary material, table

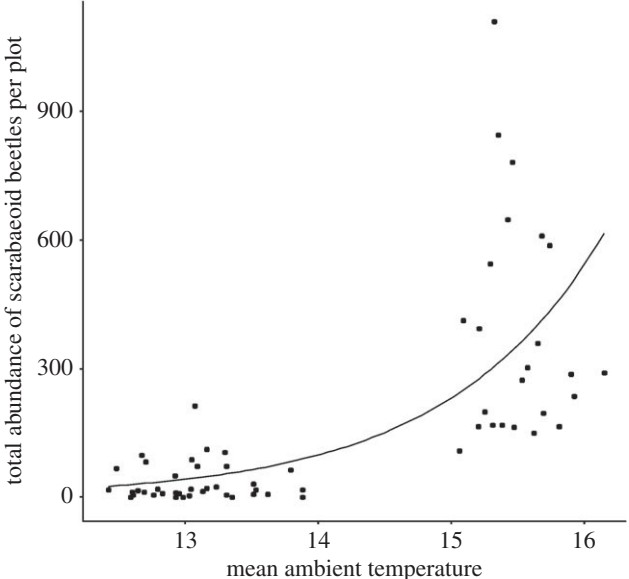

**Figure 3.** Effect of the 'mean ambient temperature' on the total abundance of scarabaeoid dung beetles per plot. Observed values (circles) and predicted values (lines) for the quasi-Poisson-GLM model: $F = 16.84$, $p < 0.001$. See electronic supplementary material, table S3$a$ for coefficients from model simplification.

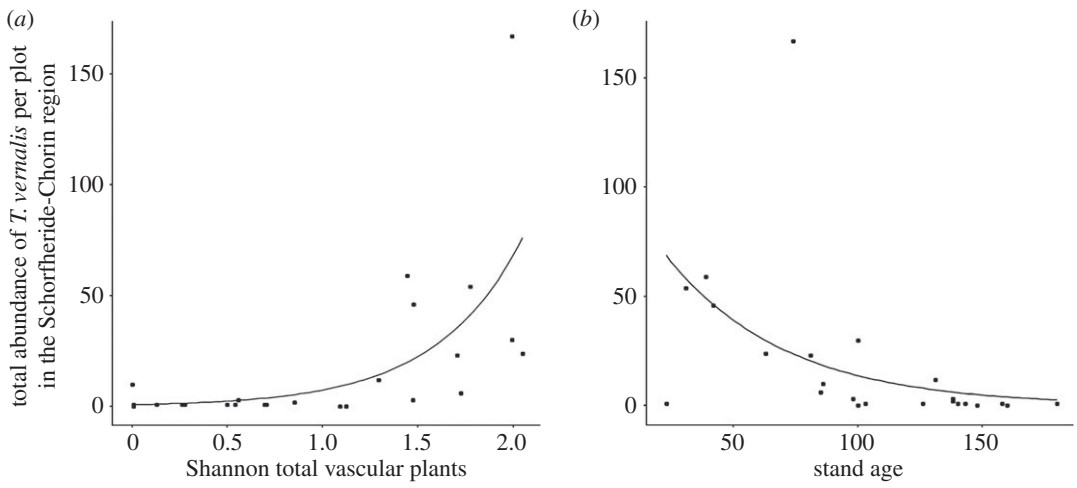

**Figure 4.** Plots depicting the relationship between the total abundance of *T. vernalis* per plot in the Schorfheide-Chorin region and (*a*) Shannon total vascular plants, (*b*) stand age; observed values (circles) and predicted values (lines) for the quasi-Poisson-GLM model ($F = 53.97$, $p < 0.001$). See electronic supplementary material, table S3$c$ for coefficients from model simplification.

S3$b$). The simplified model explained 68.6% of the variance. Forest management intensity (SMI) did not affect *A. stercorosus* abundance (quasi-Poisson-GLM: $F = 0.47$, $p = 0.496$).

## 3.5. Influence of environmental variables on the abundance of the single geotrupid species *T. vernalis* in the Schorfheide-Chorin region

The total abundance of *T. vernalis* increased with higher fine-sand contents, higher vascular plant diversity (Shannon (figure 4*a*) and Simpson's diversity) and for stand density of the main tree species (electronic supplementary material, table S1$c$); in contrast, the abundance of *T. vernalis* decreased with higher forest stand age (figure 4*b*; electronic supplementary material, table S3$c$) (quasi-Poisson-GLM: $F = 53.97$, $p < 0.001$). The simplified model explained 97.7% of the variance. Forest management intensity (SMI) did not affect *T. vernalis* abundance (quasi-Poisson-GLM: $F = 0.99$, $p = 0.334$).

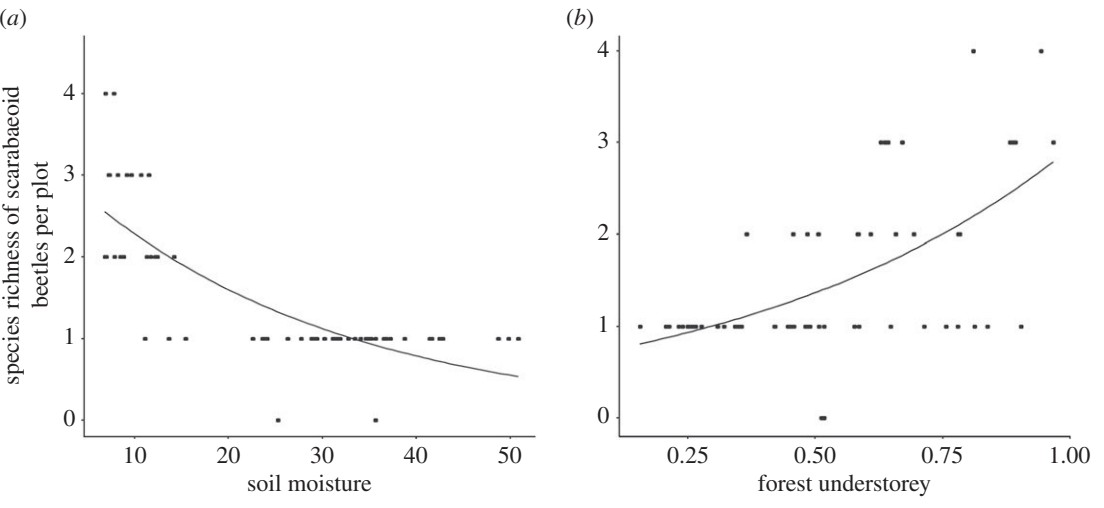

**Figure 5.** Plots depicting the relationship between species richness of scarabaeoid dung beetles per plot and (*a*) soil moisture and (*b*) forest understorey. (*a,b*) Observed values (circles) and predicted values (lines) for the quasi-Poisson-GLM model ($F =$ 28.54, $p < 0.001$). See electronic supplementary material, table S5 for coefficients from model simplification.

## 3.6. Influence of environmental variables on dung beetle species richness

Dung beetle species richness across all three regions declined with higher soil moisture (figure 5*a*) but increased with a higher proportion of forest understorey (figure 5*b*; electronic supplementary material, table S5) (quasi-Poisson-GLM: $F = 28.54$, $p < 0.001$). The simplified model explained 64.2% of the variance. Forest management intensity (SMI) did not affect the dung beetle species richness (quasi-Poisson-GLM: $F = 0.85$, $p = 0.361$).

## 3.7. Influence of environmental variables on dung beetle diversity (Simpson's dominance and Shannon's diversity) in the Schorfheide-Chorin region

Shannon's diversity of dung beetles was higher on plots with higher vascular plant Shannon's diversity (figure 6; electronic supplementary material, table S7*a*) (Gaussian-GLM: $F = 6.59$, $p = 0.004$). The simplified model predicting Shannon's diversity explained 73.3% of the variance. Concerning the sole influence of SMI, no significant difference was noted between managed and unmanaged forests in Shannon's diversity of all captured dung beetle individuals in the Schorfheide-Chorin region (Gaussian-GLM: $F = 3.57$, $p = 0.077$). In the Schorfheide-Chorin region, Simpson's dominance of dung beetles increased with higher vascular plant Shannon's diversity and higher fine-sand contents (Gaussian-GLM: $F = 11.20$, $p = 0.001$; electronic supplementary material, table S7*b*). The simplified model predicting Simpson's dominance explained 59.9% of the variance. The intensity of forest management (SMI) showed no consequence for Simpson's dominance of all trapped dung beetle individuals in the Schorfheide-Chorin region (Gaussian-GLM: $F = 1.93$, $p = 0.184$).

# 4. Discussion

Data from dung beetles captured on 61 decomposing piglets in variously managed forests were compiled in three regions of the German republic. We found that a remarkably large number of these dung beetles were attracted to the cadavers. Moreover, we could observe a large variation in abundance and species richness between experimental plots and decomposition stages. Our results revealed that, along with ambient temperature, various habitat characteristics, especially soil and tree stand parameters, and the prevailing vascular plant diversity strongly affected dung beetle abundance, diversity and species richness. The intensity of forest management had no considerable impact on the biodiversity of captured dung beetles.

## 4.1. Dung beetle abundance and richness across regions

During our sampling campaign, we captured and determined nine dung beetle species in pitfall traps surrounding piglet carcasses. The prevalent species was the geotrupid forest dung beetle

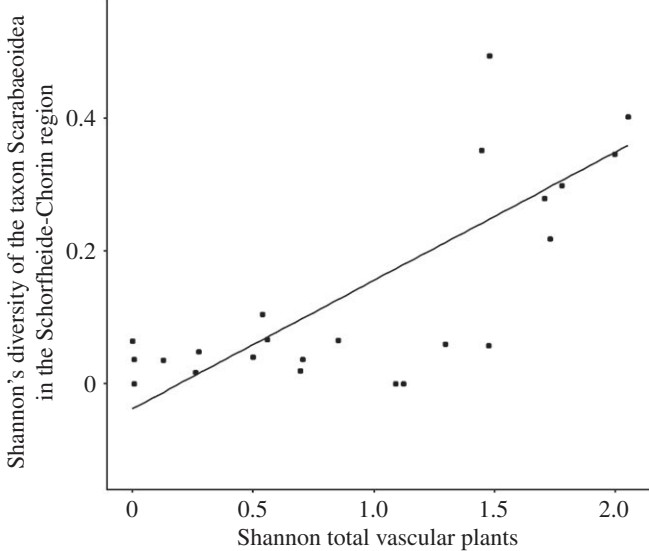

**Figure 6.** Plots depicting the relationship between Shannon's diversity of scarabaeoid dung beetles in the Schorfheide-Chorin region and Shannon's diversity of all vascular plants. Observed values (circles) and predicted values (lines) for the Gaussian-GLM model ($F = 6.59$, $p = 0.004$). See electronic supplementary material, table S7 for coefficients from model simplification.

*A. stercorosus*, followed by the geotrupid spring dung beetle *T. vernalis*. From all other taxa of Scarabaeidae and Aphodiidae, we registered only 15 specimens altogether, compared with 10 473 specimens in the Geotrupidae taxon. In a previous study, similarly high numbers of *A. stercorosus* and *T. vernalis* were caught on 18 exposed pig cadavers (mean weight = 25.8 kg) in the forests of Central Europe (western Poland) [14]. Similar to our study, *A. stercorosus* was, at 77.4%, the more abundant of the two species (in our current study: 94.9%). Frank *et al*. [27] studied land use effects on dung beetle communities in the Biodiversity Exploratories by using dung-baited pitfall traps installed at 150 forest and 150 grassland plots. Of their trapped dung-associated beetles (overall 18 780 individuals out of 33 species), 52% were individuals of the only two geotrupid species *A. stercorosus* and *T. vernalis* [27]. Consequently, based on such high trapping numbers at carrion and dung resources as well, we designate both geotrupid species as true copronecrophagous, having a strong impact on dung and carrion ecology. Interestingly, both studies dealing with exposed carrion revealed the same phenomenon: most geotrupid individuals resided in the soil beneath the pig cadavers, dug tunnels and fed directly on the cadaver substrate on the soil surface ([14]; current study: C von Hoermann 2014, personal observation). Hence, the large number of copronecrophagous geotrupid beetles might substantially contribute to carrion decomposition, speeding up the process with consequences for carrion food webs and nutrient cycling [8]. From here, throughout the entire section, our results discussion will set the principal focus on the two most abundant copronecrophagous species of Geotrupidae (Geotrupidae as part of the taxon Scarabaeoidea, the latter including a large amount of taxa that are not dung/necrophagous beetles), representing 99.86% of all specimens collected at our exposed piglet cadavers.

In our current study, the abundance of dung beetles varied considerably between trapping sites from zero individuals on two single plots in the Hainich-Dün zone to up to 1109 individuals on a single plot in the Schorfheide-Chorin region. This variation was also reflected in species richness, rising from a minimum of zero species on two specific plots in Hainich-Dün up to a maximum of five species on a single plot in the Schorfheide-Chorin. We reported similar variations in our previous study on silphid beetles visiting cadavers within the Biodiversity Exploratories [31]. Silphid species numbers rose from a minimum of one on a single forest plot in Hainich-Dün to a maximum of seven on a single forest plot in Schorfheide-Chorin [31]. However, this regional effect was not only limited to necrophagous and copronecrophagous beetles. A comprehensive survey of scarabaeoid beetles attracted to dung found that, overall, dung beetle biomass was 80 times higher in forests than in grasslands in the Schorfheide compared with multiplying factors of only '10 times' in the Alb and '20 times' in the Hainich [27]. The unequally distributed numbers of cadaver-visiting dung and silphid beetle individuals and species might be correlated with the frequency and abundance of game animals, especially red deer (*Cervus elaphus*), in forests of the German Republic. Free-ranging red deer exists

only in the region of Schorfheide-Chorin with high numbers of 40–70 specimens per 1000 ha [69]. The higher disposability of large vertebrate cadavers and deer dung in forest areas of the Schorfheide-Chorin zone might have set the stage for the establishment of higher densities of copronecrophagous dung beetle (in particular Geotrupidae) and necrophagous silphid [31] beetle populations compared with beetle communities in Hainich-Dün and the Schwäbische Alb, living in areas that are free of red deer. This assumption is substantiated by the findings of a Japanese study examining the effects of sika deer overabundance on dung and carrion beetle communities [70]. Dung beetles and carrion beetles responded positively to sika deer overabundance on Nakanoshima Island, Hokkaido, Northern Japan. Abundance, species richness and diversity were higher at the island site compared with lakeshore areas, which served as control sites [70].

Although we have found large variation in beetle abundance between areas, the high catch numbers overall confirm the previous notion that the visitation of cadaver substrate by copronecrophagous dung beetles does not occur by accident, but is rather an important component of carcass succession. Therefore, as an important next step, the possibility of competition between groups of necrophagous, copronecrophagous and coprophagous beetles has to be examined in future carrion ecology studies regarding a potential influence on the modelling outcomes. This could be done by completely comparing the results obtained from dung using Geotrupidae, Aphodiidae and Scarabaeidae [27] with respect to results obtained from necrophagous and copronecrophagous beetles like some members of the Silphidae [31] and the Geotrupidae (*A. stercorosus* and *T. vernalis*) taxon.

## 4.2. Importance of decomposition stages on dung beetle abundance

Our results demonstrate that, across all three regions, cadavers in advanced decay attracted a much higher abundance of dung beetle specimens than those that were fresh, putrefied or skeletonized. This finding highlights the importance of monitoring a cadaver-associated target taxon throughout the course of decomposition in order to avoid underestimation of abundance in later statistical modelling [71]. Our finding is also fully in agreement with the result reported by Jarmusz and Bajerlein [14], who detected an increase in *A. stercorosus* and *T. vernalis* numbers during the bloating and post-bloating stages of cadaver decomposition, with the highest peak during advanced decay. The late stages of decomposition are marked by a substantial release of cadaveric fluids into the soil. In particular, the stage of advanced decay is associated with a pronounced increase in the concentration of soil nitrogen and other nutrients such as potassium, calcium and magnesium [72]. This readily available mixture of organic matter and nutrients appears to form a highly attractive food source for certain geotrupid species.

On a proximate level, olfactory cues produced by bacterial communities [73] are generally believed to be the primary driver of cadaver colonization by arthropods. The decomposition process of mammalian cells and tissues starts soon after the death and leads to the release of volatile organic compounds (VOCs) [73,74] that attract many arthropods [75–78]. The release of VOCs is highly dynamic and depends strongly on the decomposition stage of the carcass [73,74,79]. Previous studies have revealed that dung beetles are sensitive to specific dung volatiles, such as 2-butanone, butyric acid, phenol, p-cresol, indole and skatole [80,81]. The same volatiles are also emitted by decomposing cadavers. Indeed, Stavert *et al.* [82] have detected p-cresol and phenol as the most common phenolic compounds that occur in many dung and many carrion types. Moreover, in a previous forensic chemo-ecological study, we have been able to show that the advanced decay and dry remains stages of piglet cadavers are dominated by high relative amounts of phenol [83]. Hence, phenol might be an important substance explaining the specific attraction of copronecrophagous beetles to the advanced decay stage. However, more chemo-ecological work is needed. In the framework of the current study, we have collected the VOCs released by the decomposing piglet cadavers. This, together with electrophysiological and behavioural studies, should provide a more detailed picture about those substances that drive the attraction of *A. stercorosus* and *T. vernalis* beetles to a carrion resource.

## 4.3. Effect of anthropogenic land use on dung beetle diversity and abundance

As early as 2008, Nichols *et al.* [24] reviewed the multiple lines of evidence from tropical and temperate systems indicating that changes in anthropogenic land use and mammal fauna on a local and regional scale have the ability severely to alter the patterns of dung beetle abundance and species diversity. Moreover, Barragán *et al.* [30] documented a decline in the functional biodiversity of copronecrophagous beetle communities, resulting from changes in human land use. More recently, Frank *et al.* [27] have

shown that land use affects dung beetle communities and their ecosystem service in forests and grasslands within the Biodiversity Exploratories. In our study, we have not been able to verify this effect of anthropogenic land use on the abundance of dung beetles attracted to decaying cadavers. However, in the Schorfheide-Chorin region, in which we trapped the highest number of beetles, we found a tendential impact of forest management intensity on the diversity of dung beetles attracted to a carrion resource.

Surprisingly, higher forest management intensity increased the dung beetle diversity. This finding was further supported by the finding that forests with a low standard deviation of the tree diameter at breast height, a typical indicator of monoculture stands, contained the highest dung beetle diversity. Although our result here might be counterintuitive at first sight, it corresponds to the observation made by Frank et al. [27] who revealed an increase in dung removal activity of +22.3% from forests with no non-native trees to monospecific conifer stands inside the Biodiversity Exploratories. A possible explanation of their and our data might be that conifer stands are particularly attractive for game species (dung and cadaver suppliers) because these habitats are more sheltered and have a strong development of understorey vegetation serving as food for red deer [84,85].

## 4.4. Effects of other environmental parameters on dung beetle diversity and abundance

In addition to forest management intensity, our study indicated that other environmental parameters influenced the abundance of dung beetles and their diversity. In particular, soil characteristics such as fine-sand content and soil moisture had an impact. A higher fine-sand extent in forest soils had a positive impact on the numbers of A. stercorosus and T. vernalis and also raised the Simpson's dominance of dung beetles. Soil moisture, on the other hand, had a negative effect on species richness. For the two most abundant species in our study, high soil moisture negatively affects larval development [14,36]. As A. stercorosus and T. vernalis are geotrupid-tunnellers (shown tunnelling activity with very short dung removal times in the family of Geotrupidae and the genus Onthophagus in the Biodiversity Exploratories [27]) and primarily bury dung, we assume that loose sandy and arid soils present suitable living habits for their digging behaviour (less energetically costly) and consequently cause higher dung beetle abundance over time.

Moreover, forest characteristics, such as stand age and vascular plant diversity, had an effect on the abundance of copronecrophagous dung beetle. Our analyses revealed a significantly higher abundance of T. vernalis on above-ground exposed piglet cadavers in younger than in older stands. A similar negative correlation between the numbers of T. vernalis and stand age was observed in a study involving the capture of dung beetles in unbaited Barber traps in the Wipsowo Forest District in Poland. This study found that numbers of T. vernalis gradually decreased with increasing stand age in a clear-cut area [86]. The reason for the findings in both studies could be the absence of undergrowth vegetation in older stands (completely shaded ground, scant litter consisting of pine needles [86]). The absence of underground vegetation, as mentioned above, is a negative precondition for large herbivores, such as roe and red deer, and consequently for the deposition and availability of their faeces and their incoming cadavers.

The abundance of the geotrupid species T. vernalis on exposed piglet cadavers was positively correlated with a high vascular plant diversity on the respective forest plot. Interestingly, we also found a higher dung beetle diversity and Simpson's dominance on forest plots with higher vascular plant diversity. This finding correlates with observations that have been made in herbivorous insects, such as butterflies, and that show that flower abundance, understorey herb cover and vegetation diversity promote butterfly diversity [87–90]. Thus, the maintenance of native understorey vegetation through adequate forestry practices has been suggested to conserve butterflies in heavily managed conifer plantations [89–91]. Our study indicates that this might also hold for copronecrophagous dung beetles.

However, the positive effect of vascular plant diversity on dung beetle diversity and abundance might not be unidirectional. The secondary seed dispersal provided by tunnelling geotrupid species, such as T. vernalis, might, in turn, have a positive effect on vascular plant diversity. In burying dung containing plant seeds, dung beetles relocate seeds both horizontally (roller species) and vertically (tunnelling species) [23]. Secondary seed dispersal is believed to play an important role in plant recruitment [92] by directing dispersal to favourable microclimates for germination and emergence [93] and by decreasing residual post-dispersal seed clumping [94]. We consider that these advantages in seed dispersal provided by copronecrophagous dung beetles contribute to the high diversity of vascular plants that is found at former cadaver sites, months after the last decomposing remains have disappeared (C von Hoermann 2019, personal observation). Further studies have been initiated to confirm the relationship between vascular plant and dung beetle diversity and to analyse the nature of this relationship in more detail.

## 4.5. Influence of climatic conditions on dung beetle abundance and diversity

In a previous study examining cadaver-visiting silphid beetles within the Biodiversity Exploratories, we found that higher temperatures in the surroundings clearly diminish Simpson's dominance in this taxon [31]. We have suggested that this reduction of the Simpson's dominance in predacious and/or carrion-eating silphid beetles alters the calculable succession pattern of carrion insects [9] and the whole decomposition course and, therefore, the nutrient recycling in ecosystems [95]. In the present study, we have found no effect of temperature on Simpson's dominance and biodiversity in copronecrophagous dung beetles. However, higher ambient temperatures significantly raise the abundance of dung beetles on above-ground exposed piglet cadavers. We assume that an anthropogenic induced disturbance such as monocultured pine forest habitats in Central Europe, showing typically high air temperatures near the sandy forest floor [14], results in higher abundance values of poikilothermic copronecrophagous dung beetles on cadaveric resources. Therefore, even within monocultured forest stands for timber, such high abundance values of copronecrophagous dung beetles can ensure a valuable ecosystem service by a rapid cadaveric nutrient return into the ecosystem. This underlines the plasticity of dung beetles towards anthropogenic disturbances and makes them particularly worthy of protection.

## 5. Conclusion

In our study, large numbers of dung beetles were attracted to all of our 61 piglet cadavers. Thus, dung beetles, in particular the most abundant geotrupid species *A. stercorosus* and *T. vernalis*, caught on animal carrion can be no longer considered mere by-catch. Noticeably, the two geotrupid taxa *A. stercorosus* and *T. vernalis* represented 99.9% of all collected cadaver-visiting dung beetle specimens compared with only 0.1% represented by the taxa Aphodiidae and Scarabaeidae. With regard to successional investigations, dung beetles attracted to vertebrate carcasses showed their highest abundance in later decay, more specifically in advanced decay stage. The latter provides a readily available mixture of organic matter and nutrients, representing a highly attractive food source for numerous copronecrophagous geotrupid individuals. Consequently, dung beetles might represent an essential factor not only for rapid faecal but also for cadaveric recycling in intact ecosystems. Their contribution to carrion decomposition might have a great impact on nutrient cycling and carrion food webs. Nevertheless, the role of cadaver resources in the biology of dung beetles prompts many questions that should be addressed by future research projects on dung beetles and/or carrion ecology. For instance, do adults of certain dung beetle species depend on cadavers as a food source or is their interest in cadavers plain opportunism? In that context, it could be examined if dependency on cadavers or plain opportunism results in a quantifiable effect of competition between Silphidae, Geotrupidae, Scarabaeidae and Aphodiidae for the dung and for the carrion resource, altering their abundance, species richness and diversity. It would be also important to know how our findings obtained in summer vary throughout the seasons. Furthermore, research into whether dung beetles use the same volatiles to locate both dung and cadavers and whether they are able to discriminate these potential food sources by smell would be of interest.

With regard to biodiversity investigations, our study clearly shows that anthropogenic presentation of cadavers is a useful way to examine dung beetle populations. Following the baiting of dung beetles, we have found large variations in abundance and species richness between the various forest habitats in three regions of Germany. We have demonstrated that soil characteristics, forest stand, forest understorey, vascular plant diversity and temperature are important factors determining the abundance and diversity of copronecrophagous dung beetles. Our results designate the sandy soils typically present in arid pine forests as a suitable habitat for high dung beetle abundance and diversity. In our study, human forest management has been revealed to have no negative effect on dung beetle populations in the explored areas. On the contrary, dung beetle diversity is highest in forests with a low standard deviation of tree diameter, a typical indicator of monoculture stands. However, the monospecific conifer stands included in our study offer one of the most sheltered habitats and have a strong development of understorey vegetation, a particularly attractive habitat for dung and cadaver suppliers such as red deer. Thus, as long as diverse and lush understorey vegetation is present, dung beetles thrive, including in warm, dry and highly used monocultured forest stands. This underlines the plasticity of copronecrophagous members of dung beetles towards anthropogenic disturbances and makes them particularly worthy of protection, as they offer valuable ecosystem services by recycling nutrients, even in disturbed ecosystems.

Ethics. No animals were killed for this study. All cadavers of exclusively stillborn piglets were obtained under veterinary supervision (special permit for animal by-products (EG) No. 1069/2009) from a local pig farmer (Winfried Walter, Gögglingen, Germany). Fieldwork permits were issued by the responsible state environmental offices of Baden-Württemberg, Thüringen and Brandenburg (according to § 72 BbgNatSchG). For the field sampling of arthropods, an exemption existed concerning § 67 BNatSchG and species protection legislation according to § 45 BNatSchG.

Data accessibility. All relevant data supporting the underlying conclusions of our research are accessible from the Dryad Digital Repository: https://dx.doi.org/10.5061/dryad.1ns1rn8ps [47] and from the BExIS platform (Biodiversity Exploratories Information System, https://www.bexis.uni-jena.de; details on the identity of datasets included, see electronic supplementary material, table S1).

Authors' contributions. C.v.H. conceived and designed the study and was responsible for fieldwork and data acquisition, funding acquisition, statistical analyses and drafting the manuscript; S.W. participated in fieldwork and, together with M.D., was also responsible for data acquisition, i.e. species determination; M.A. and S.S. conceived, designed and coordinated the study. Together with C.v.H., they were responsible for funding acquisition and helped to draft the manuscript. All authors gave final approval for publication.

Competing interests. We have no competing interests.

Funding. This work was funded by the German Research Foundation (DFG) Priority Program 1374 'Infrastructure-Biodiversity-Exploratories' (AY 12/9-1, STE 1874/4-1 to C.v.H.; http://www.biodiversity-exploratories.de).

Acknowledgements. We thank the managers of the three Exploratories, namely Kirsten Reichel-Jung, Swen Renner, Katrin Lorenzen, Sonja Gockel, Kerstin Wiesner and Martin Gorke, for their work in maintaining the plot and project infrastructure; Ralf Schlehahn and Ulf Pommer of the Local Management Team Schorfheide-Chorin for intensive fieldwork assistance; the student assistants Katrin Brodbeck, Dorothee Wilhelm, Adrian Monteleone, Linda Stabenau, Annika Wess, Andreas Fischer, Ulrich Neumüller, Julia Thomas, Kevin Kröger and Stefanie Kern for their great enthusiasm during fieldwork; Ann-Marie Rottler-Hoermann for essential logistic management of the fieldwork and helpful comments on the manuscript; Theresa Jones for linguistic advice; Winfried Walter for kindly providing stillborn piglets; Christian Fischäß for veterinarian support; Thomas Ley as a representative of the veterinarian authority; Christiane Fischer and Simone Pfeiffer for giving support through the central office; Michael Owonibi for managing the central data base; and Markus Fischer, Eduard Linsenmair, Dominik Hessenmöller, Jens Nieschulze, Daniel Prati, Ingo Schöning, François Buscot, Ernst-Detlef Schulze, Wolfgang W. Weisser and the late Elisabeth Kalko for their role in setting up the Biodiversity Exploratories project. Acknowledgements are determined by rules of the Biodiversity Exploratories.

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
