## [Reviewer comments · Royal Society Open Science]

Review History

RSOS-191722.R0 (Original submission)

Review form: Reviewer 1 (A. Gómez-Cifuentes)

Is the manuscript scientifically sound in its present form?

Yes

Are the interpretations and conclusions justified by the results?

Yes

Is the language acceptable?

Yes

Do you have any ethical concerns with this paper?

No

Have you any concerns about statistical analyses in this paper?

No

Recommendation?

Accept with minor revision (please list in comments)

Comments to the Author(s)

Dear authors;

The manuscript focus on a very interesting topic, which is the influence of abiotic and biotic factors in the diversity of copronecrophagous beetles. Also, the study focus on several important topics associated to ecosystem process such as the carrion decomposition, which is unusual because most of studies with Scarabaeidae beetles evaluated the dung decomposition. Data and analysis performed by authors seems to be the appropriate for this kind of studies. After I read the manuscript, there are some major issues that should be addressed before this manuscript can be published:

Major comments

1. Authors should include the aim and hypothesis of your work.
2. I consider that table 1, table 2, table 3 and figure 1 could be in electronic appendix. Besides, I suggest that figure 3b and 3c, and figure 5d do not be published because these values are nor statistical significance ($p > 0.05$).
3. I think could be interesting if authors include more information about the ecology/biology of these two main species they found, which seems the dominant species in central region of Europe.
4. Be careful with the use of not statistical significance values ($p > 0.05$); I understand that is very common the use of that information as "tendency" o "patterns".

Recommended literature.

Dortel, E., Thuiller, W., Lobo, J. M., Bohbot, H., Lumaret, J. P. & Jay-Robert, P. (2013). Potential effects of climate change on the distribution of Scarabaeidae dung beetles in Western Europe. *Journal of Insect Conservation*, 17, 1059-1070. <https://doi.org/10.1007/s10841-013-9590-8>

Giménez, V. C., Verdú, J. R., Guerra Alonso, C. B. & Zurita, G. A. (2018). Relationship between land uses and diversity of dung beetles (Coleoptera: Scarabaeinae) in the southern Atlantic forest of Argentina: which are the key factors? *Biodiversity and Conservation*, 27, 3201-3213. <https://doi.org/10.1007/s10531-018-1597-8>

Neita, J. C. & Escobar, F. (2012). The potential value of agroforestry to dung beetle diversity in the wet tropical forests of the Pacific lowlands of Colombia. *Agroforestry Systems*, 85, 121-131. <https://doi.org/10.1007/s10457-011-9445-9>

Verdú, J. R. & Lobo, J. (2008). Ecophysiology of thermoregulation in endothermic dung beetles: Ecological and geographical implications. *Insect Ecology and Conservation*, 661, 1-28. ISBN: 978-81-308-0297-8

Minor comments

Abstract

Lines 30-31: I think this is wrong. The highest beetle diversity was associated to Shannon total vascular plants (both, Shannon and Simpson) and fine sands (only Simpson); whereas total abundance was associated to mean ambient temperature (see Table 1 and 3). However, the last result is very important because copronecrophagous beetles are highly sensitive to changes in temperature and humidity, due to their physiological restrictions (Neita and Escobar, 2012; Verdú and Lobo, 2008; Dortel et al., 2013; Giménez Gómez et al., 2018).

Introduction

Lines 83-86: this is the aim of your study? This seems like a short description of the activities you made. I suggest that rephrased this statement, try to be more specific about what was the aim of this study and included the hypothesis

Lines 86-87: this should be in the results section

Lines 87-88: this should be in the methods section

Lines 88-89: this should be in the discussion/conclusions section

Methods

Lines 101-102: Your study area were the Biodiversity Exploratories, but, you compared three regions; so, I suggest that changed “study area” to “region” from here to the end of this section

Results

Lines 178-179: I suggest “we captured 10488 individuals from the following nine scarabaeoid beetles species”; or you can use a similar statement of lines 182-183

Line 189: I suggest that change “importance” to “effects”

Lines 196-197, 217-218, 229-230, 240-241: Avoid “introduction statements” and just go directly with your results

Lines 199-200: I do not agree with authors. I suggest that analyzed your results if it has statistical differences ($p < 0.05$). Besides, the influence of mean ambient temperature is a great results because copronecrophagous beetles are highly sensitive to changes in temperature and humidity, due to their physiological restrictions (Neita and Escobar, 2012; Verdú and Lobo, 2008; Dortel et al., 2013; Giménez Gómez et al., 2018)

Lines 200-201: I suggest that unify this statement in lines 198-199

Lines 201-202: Again, I do not agree with authors. Soil moisture had not effect on overall beetle abundance ($p = 0.08$)

Lines 210-212: I found this statement redundant. Authors should simplify this

Lines 220-221: I do not agree with authors ($p > 0.05$)

Lines 217-224: This is very redundant and complex. Authors should simplify this as “The total abundance of *T. vernalis* increased with higher fine-sand contents, higher vascular plant diversity (Shannon and Simpson’s diversity) and for stand density of the main tree species (table 1c); in contrast, the abundance of *T. vernalis* decreased with higher forest stand-age (figure 4b, table 1c) (quasi-Poisson-GLM: $F = 53.97$, $p < 0.001$).”

Line 225: I do not agree with authors ($p > 0.05$)

Lines 232-233: I do not agree with authors ($p = 0.06$)

Lines 229-234: I think this is a redundant statement

Line 235: I do not agree with authors ($p = 0.06$)

Lines 243-244: I do not agree with authors ($p = 0.07$)

Lines 245-246: I do not agree with authors ($p = 0.07$)

Lines 240-246: Authors should rephrased this statement because it is redundant

Lines 247-250: I do not agree with authors ($p = 0.08$)

Lines 252-253: Authors should unify this lines

Discussion

Line 268: ScarAbaeoid

Review form: Reviewer 2

Is the manuscript scientifically sound in its present form?

No

Are the interpretations and conclusions justified by the results?

No

Is the language acceptable?

No

Do you have any ethical concerns with this paper?

No

Have you any concerns about statistical analyses in this paper?

No

Recommendation?

Major revision is needed (please make suggestions in comments)

Comments to the Author(s)

Review of RSOS-191722 "Forest habitat parameters influence abundance and diversity of cadaver-visiting dung beetles".

LL83-89. The questions being asked are not clear. Although the broad aim is given, I would like to see a list of questions that are answered in this study.

L116. please provide some details on the pitfall trapping method. how long were they opened for? what preservative did you use? what diameter? these details should be in this paper.

L126. Some further detail is needed here, without reference to the supporting material. For example, were variables collected at each carcass? or were they averaged across each 100 x 100 plot? Why were so many variables examined? There seems no clear link between the variables and expected dung beetle abundances/occurrences.

L140. GLMs are univariate approaches (you are investigating predictors of individual response variables).

L151. what trap types? I can only see pitfall traps were used, please clarify.

L164. What is the question being addressed here? This approach seems very much a fishing exercise, with all sorts of variables examined to find a potential relationship. I think that a revision of the methods would be very useful to explain your choice of variables, and how this relates to your questions.

L169. this paragraph seems added at the last minute and doesn't quite fit here. these response variables would be better described earlier at the beginning of the stats section.

L189. here (and elsewhere) consider replacing 'degradation phase' with 'decay stage', as this is more commonly used and understood.

L200. what does 'tendentally guilty' mean? perhaps a translation issue.

General comment: I wonder how mean temperature changed with decay stage? If days were warmer, this might also explain higher abundance during later decay.

LL196-256. Results describing environmental effects on beetle diversity and abundance of individual species could be reduced in length. It is currently a bit repetitive, and the authors might consider looking for ways to shorten this section to highlight only the key results.

L259. unclear what this 'framework' is. Do you mean your study design?

L332. I am still unsure what the land uses are. In your methods you say that all sites were subject to forestry management, so what are the measures of land use and intensity? These variables are distinct from local habitat/environmental variables you examined, and so should be a separate question, yes?

LL411-435. I think that the conclusions do not summarise the key findings very well. What is new and different about his study that no one else has described? It is not surprising that you found lots of scarabs at pig carcasses. However, I think it is interesting they had highest abundance in

later decay, and that some species were more abundant than others. Clearly describing the main results will help to the reader to see how your study has added to the literature.

Review form: Reviewer 3 (Juan Márquez)

Is the manuscript scientifically sound in its present form?

Yes

Are the interpretations and conclusions justified by the results?

Yes

Is the language acceptable?

Yes

Do you have any ethical concerns with this paper?

Yes

Have you any concerns about statistical analyses in this paper?

No

Recommendation?

Accept with minor revision (please list in comments)

Comments to the Author(s)

1. Title: I suggest being specific: "in Germany", because the results could not be the same in other parts of the World.

2. Throughout the entire document: I think the paper can focus only on the two species of Geotrupidae, because the species collected from Aphodiidae and Scarabaeidae (Scarabaeinae) are represented by very few specimens (1 to 4), while in the study of Frank et al. (2017) (done in the same area), these same species were collected with a notable greater abundance, which seems to indicate that they are coprophagous (aspect that I have also read in several works) and their collection in carrion is occasional. Unless the authors provide convincing information that these species are copro-necrophagous or important reasons not to exclude them from the study.

The seven collected species of Aphodiidae and Scarabaeidae (Scarabaeinae) represent only 0.14% of the 10,488 specimens collected and the 99.86% correspond to the two geotrupid species.

Species collected on carrion of Scarabaeidae and Aphodiidae are included and analyzed by Frank et al. (2017) in their study of dung beetles associated to dung as part of the same big German project. Also, the two geotrupid species are included in that study, but their abundances were similar to the abundances reported in the present manuscript, which is not discussed, but I think is an evidence that these species are in fact copro-necrophagous, and their analyses, as part of dung beetles associated to dung and carrion, is important, because they are two of the most abundant species with a strong impact in the dung and carrion ecology.

If the study focuses only on Geotrupidae, the confusion I noticed throughout the text will be avoided, where Scarabaeoidea is referred to as equivalent to dung beetles; however, Scarabaeoidea includes a large amount of taxa, many of which are not dung beetles.

3. Introduction, page 2, lines 52-53: "During the 52 so-called 'post-bloating stage' of cadaver decomposition, the large feeding masses of fly maggots are joined by 53 predatory coleopterans, i.e. by members of the Silphidae, Staphylinidae and Histeridae." The majority of the silphid species are necrophagous, I think the word "some members of" can help (not all staphylinid species are predatory, but yes the majority).

Page 2, line 62: If it is pertinent to the journal, the author and year of each species could be added only the first occasion that is written.

Page 3, lines 88-89: "Our results can be generalized, as our study encompasses three regions differing in their geology, topography and climate." This statement could be debatable, because the biogeographic history is not the same for any place in the World. Maybe "Our result can be generalized in Germany" or "in Central Europe".

4. Methods and Materials, page 4, line 127: "...important factors that affect the abundance of copro-necrophagous dung beetles." As commented previously, the Aphodiidae and Scarabaeidae species collected in this work probably do not are copro-necrophagous.

5. Results, page 6, lines 208 and 216: "Influence of environmental variables on the abundance of the single scarabaeoid species *A. stercorosus*", and "Influence of environmental variables on the abundance of the single scarabaeoid species *T. vernalis* in the Schorfheide-Chorin region", change "scarabaeoid" for "geotrupid".

6. Discussion, page 7, lines 271 and 272: "From all other taxa of Scarabaeidae and Aphodiidae, we registered only 15 specimens altogether, compared with 10473 specimens in the Geotrupidae taxon." Or it may be that these particular species are not carrion eaters but of dung. Here you probably need compare the abundance of this species with the results of Frank et al. (2017). The great abundance of the two geotrupid species also is reported in that paper (coinciding with the copro-necrophagous habits).

Page 7, line 278: "Hence, the large number of coprophagous dung beetles might substantially contribute to carrion decomposition...". Coprophagous or copro-necrophagous? I have the impression that there is no distinction between one eating habit and another.

Page 7, lines 284-285: "We reported similar variations in our previous study on silphid beetles visiting cadavers within the Biodiversity Exploratories." The authors have previously studied the silphids, and in this contribution Geotrupidae (and probably some Aphodiidae and scarabs) associated to carrion; other colleagues have studied the Aphodiidae, Scarabaeidae and Geotrupidae using dung. I have two observation in relation with this magnificent knowledge: do not consider the fact of competition between these groups affecting the results (at least their consideration to future studies) and do not integrate completely (or compare) the results of Geotrupidae, Aphodiidae and Scarabaeidae obtained using dung with respect to results using carrion.

Page 8, lines 287-289: "A comprehensive survey of scarabaieoid beetles attracted to dung found that, overall, dung beetle biomass was 80 times higher in the Schorfheide than in the two other regions [23]." I am not sure if Frank et al. (2017) say the same that you say. I cannot access to the paper (would have to buy it), but yes at the Ph. D. thesis of Frank, and I am not sure that the text are the same in the thesis and the paper, but I think yes. In the thesis is written "Overall, dung beetle biomass was 10 times higher in forests than in grasslands in the Alb (Welch t-test, $t = 3.83$, $p < 0.001$), 20 times in the Hainich ($t = 5.62$, $p < 0.001$), up to 80 times in the Schorfheide ($t = 8.99$, $p < 0.001$) (Fig. 2b)." (Chapter 2, page 20). The comparison is between forest and grasslands in each site, not between sites.

Page 9, lines 359-360: "As *A. stercorosus* and *T. vernalis* are tunnellers and primarily bury dung [78], ...". This reference (Anduaga, 2004) do not speak about these species, because focused on dung beetles from a place of Durango, Mexico, and this geotrupid species do not exist in Mexico.

Page 10, lines 396-397: "We have suggested that this reduction of the Simpson's dominance in predacious silphid beetles alters the calculable...". All studied silphid species are predaceous? I think not, I think the majority are carrion eating.

7. Conclusions, page 11, lines 411-412: "Thus, scarabaeoid beetles caught on animal carrion can be no longer considered mere by-catch." This is true for the two geotrupid species, but I think not for any Scarabaeoidea species; therefore, not for any Scarabaeidae and Aphodiidae species.

Page 11, lines 416-417: "For instance, do adults of certain dung beetle species depend on cadavers as a food source or is their interest in cadavers plain opportunism?" What response can suggest the fact that the abundance of the two geotrupid species in dung (Frank et al. 2017) and carrion

are near similar? Maybe it is a good place to do a question about the effect of competence between Silphidae, Geotrupidae, Scarabaeidae and Aphodiidae for the dung and for the carrion in their abundance, richness and diversity. Another question: how do found results vary throughout the seasons?

Decision letter (RSOS-191722.R0)

17-Dec-2019

Dear Dr von Hoermann,

The editors assigned to your paper ("Forest habitat parameters influence abundance and diversity of cadaver-visiting dung beetles") have now received comments from reviewers. We would like you to revise your paper in accordance with the referee and Associate Editor suggestions which can be found below (not including confidential reports to the Editor). Please note this decision does not guarantee eventual acceptance.

Please submit a copy of your revised paper before 09-Jan-2020. Please note that the revision deadline will expire at 00.00am on this date. If we do not hear from you within this time then it will be assumed that the paper has been withdrawn. In exceptional circumstances, extensions may be possible if agreed with the Editorial Office in advance. We do not allow multiple rounds of revision so we urge you to make every effort to fully address all of the comments at this stage. If deemed necessary by the Editors, your manuscript will be sent back to one or more of the original reviewers for assessment. If the original reviewers are not available, we may invite new reviewers.

- Data accessibility

It is a condition of publication that all supporting data are made available either as supplementary information or preferably in a suitable permanent repository. The data accessibility section should state where the article's supporting data can be accessed. This section should also include details, where possible of where to access other relevant research materials

such as statistical tools, protocols, software etc can be accessed. If the data have been deposited in an external repository this section should list the database, accession number and link to the DOI for all data from the article that have been made publicly available. Data sets that have been deposited in an external repository and have a DOI should also be appropriately cited in the manuscript and included in the reference list.

If you wish to submit your supporting data or code to Dryad (<http://datadryad.org/>), or modify your current submission to dryad, please use the following link:
<http://datadryad.org/submit?journalID=RSOS&manu=RSOS-191722>

- **Competing interests**

- **Authors' contributions**

- **Acknowledgements**

- **Funding statement**

on behalf of Prof Kevin Padian (Subject Editor)
openscience@royalsociety.org

Associate Editor's comments:

A trio of reviewers have kindly commented on your paper, providing a range of suggestions and requests for improvements to be made. Please can you ensure that you carefully consider their queries, incorporate necessary changes, and make a full point-by-point response to the queries if you choose to resubmit. Thank you for considering Royal Society Open Science and good luck.

Comments to Author:

Reviewers' Comments to Author:

Reviewer: 1

Comments to the Author(s)

Dear authors;

The manuscript focus on a very interesting topic, which is the influence of abiotic and biotic factors in the diversity of copronecrophagous beetles. Also, the study focus on several important topics associated to ecosystem process such as the carrion decomposition, which is unusual because most of studies with Scarabaeidae beetles evaluated the dung decomposition. Data and analysis performed by authors seems to be the appropriate for this kind of studies. After I read the manuscript, there are some major issues that should be addressed before this manuscript can be published:

Major comments

1. Authors should include the aim and hypothesis of your work.
2. I consider that table 1, table 2, table 3 and figure 1 could be in electronic appendix. Besides, I suggest that figure 3b and 3c, and figure 5d do not be published because these values are nor statistical significance ($p > 0.05$).
3. I think could be interesting if authors include more information about the ecology/biology of these two main species they found, which seems the dominant species in central region of Europe.
4. Be careful with the use of not statistical significance values ($p > 0.05$); I understand that is very common the use of that information as "tendency" o "patterns".

Recommended literature.

- Dortel, E., Thuiller, W., Lobo, J. M., Bohbot, H., Lumaret, J. P. & Jay-Robert, P. (2013). Potential effects of climate change on the distribution of Scarabaeidae dung beetles in Western Europe. *Journal of Insect Conservation*, 17, 1059-1070. <https://doi.org/10.1007/s10841-013-9590-8>
- Giménez, V. C., Verdú, J. R., Guerra Alonso, C. B. & Zurita, G. A. (2018). Relationship between land uses and diversity of dung beetles (Coleoptera: Scarabaeinae) in the southern Atlantic forest of Argentina: which are the key factors? *Biodiversity and Conservation*, 27, 3201-3213. <https://doi.org/10.1007/s10531-018-1597-8>
- Neita, J. C. & Escobar, F. (2012). The potential value of agroforestry to dung beetle diversity in the wet tropical forests of the Pacific lowlands of Colombia. *Agroforestry Systems*, 85, 121-131. <https://doi.org/10.1007/s10457-011-9445-9>
- Verdú, J. R. & Lobo, J. (2008). Ecophysiology of thermoregulation in endothermic dung beetles: Ecological and geographical implications. *Insect Ecology and Conservation*, 661, 1-28. ISBN: 978-81-308-0297-8

Minor comments

Abstract

Lines 30-31: I think this is wrong. The highest beetle diversity was associated to Shannon total vascular plants (both, Shannon and Simpson) and fine sands (only Simpson); whereas total abundance was associated to mean ambient temperature (see Table 1 and 3). However, the last result is very important because copronecrophagous beetles are highly sensitive to changes in temperature and humidity, due to their physiological restrictions (Neita and Escobar, 2012; Verdú and Lobo, 2008; Dortel et al., 2013; Giménez Gómez et al., 2018).

Introduction

Lines 83-86: this is the aim of your study? This seems like a short description of the activities you made. I suggest that rephrased this statement, try to be more specific about what was the aim of this study and included the hypothesis

Lines 86-87: this should be in the results section

Lines 87-88: this should be in the methods section

Lines 88-89: this should be in the discussion/conclusions section

Methods

Lines 101-102: Your study area were the Biodiversity Exploratories, but, you compared three regions; so, I suggest that changed "study area" to "region" from here to the end of this section

Results

Lines 178-179: I suggest "we captured 10488 individuals from the following nine scarabaeoid beetles species"; or you can use a similar statement of lines 182-183

Line 189: I suggest that change "importance" to "effects"

Lines 196-197, 217-218, 229-230, 240-241: Avoid "introduction statements" and just go directly with your results

Lines 199-200: I do not agree with authors. I suggest that analyzed your results if it has statistical differences ($p < 0.05$). Besides, the influence of mean ambient temperature is a great results because copronecrophagous beetles are highly sensitive to changes in temperature and humidity, due to their physiological restrictions (Neita and Escobar, 2012; Verdú and Lobo, 2008; Dortel et al., 2013; Giménez Gómez et al., 2018)

Lines 200-201: I suggest that unify this statement in lines 198-199

Lines 201-202: Again, I do not agree with authors. Soil moisture had not effect on overall beetle abundance ($p = 0.08$)

Lines 210-212: I found this statement redundant. Authors should simplify this

Lines 220-221: I do not agree with authors ($p > 0.05$)

Lines 217-224: This is very redundant and complex. Authors should simplify this as "The total abundance of *T. vernalis* increased with higher fine-sand contents, higher vascular plant diversity (Shannon and Simpson's diversity) and for stand density of the main tree species (table 1c); in contrast, the abundance of *T. vernalis* decreased with higher forest stand-age (figure 4b, table 1c) (quasi-Poisson-GLM: $F = 53.97$, $p < 0.001$)."

Line 225: I do not agree with authors ($p > 0.05$)

Lines 232-233: I do not agree with authors ($p = 0.06$)

Lines 229-234: I think this is a redundant statement

Line 235: I do not agree with authors ($p = 0.06$)

Lines 243-244: I do not agree with authors ($p = 0.07$)

Lines 245-246: I do not agree with authors ($p = 0.07$)

Lines 240-246: Authors should rephrased this statement because it is redundant

Lines 247-250: I do not agree with authors ($p = 0.08$)

Lines 252-253: Authors should unify this lines

Discussion

Line 268: ScarAbaeoid

Reviewer: 2

Comments to the Author(s)

Review of RSOS-191722 "Forest habitat parameters influence abundance and diversity of cadaver-visiting dung beetles".

LL83-89. The questions being asked are not clear. Although the broad aim is given, I would like to see a list of questions that are answered in this study.

L116. please provide some details on the pitfall trapping method. how long were they opened for? what preservative did you use? what diameter? these details should be in this paper.

L126. Some further detail is needed here, without reference to the supporting material. For example, were variables collected at each carcass? or were they averaged across each 100 x 100 plot? Why were so many variables examined? There seems no clear link between the variables and expected dung beetle abundances/occurrences.

L140. GLMs are univariate approaches (you are investigating predictors of individual response variables).

L151. what trap types? I can only see pitfall traps were used, please clarify.

L164. What is the question being addressed here? This approach seems very much a fishing exercise, with all sorts of variables examined to find a potential relationship. I think that a revision of the methods would be very useful to explain your choice of variables, and how this relates to your questions.

L169. this paragraph seems added at the last minute and doesn't quite fit here. these response variables would be better described earlier at the beginning of the stats section.

L189. here (and elsewhere) consider replacing 'degradation phase' with 'decay stage', as this is more commonly used and understood.

L200. what does 'tendentally guilty' mean? perhaps a translation issue.

General comment: I wonder how mean temperature changed with decay stage? If days were warmer, this might also explain higher abundance during later decay.

LL196-256. Results describing environmental effects on beetle diversity and abundance of individual species could be reduced in length. It is currently a bit repetitive, and the authors might consider looking for ways to shorten this section to highlight only the key results.

L259. unclear what this 'framework' is. Do you mean your study design?

L332. I am still unsure what the land uses are. In your methods you say that all sites were subject to forestry management, so what are the measures of land use and intensity? These variables are distinct from local habitat/environmental variables you examined, and so should be a separate question, yes?

LL411-435. I think that the conclusions do not summarise the key findings very well. What is new and different about his study that no one else has described? It is not surprising that you found lots of scarabs at pig carcasses. However, I think it is interesting they had highest abundance in later decay, and that some species were more abundant than others. Clearly describing the main results will help to the reader to see how your study has added to the literature.

Reviewer: 3

Comments to the Author(s)

1. Title: I suggest being specific: "in Germany", because the results could not be the same in other parts of the World.

2. Throughout the entire document: I think the paper can focus only on the two species of Geotrupidae, because the species collected from Aphodiidae and Scarabaeidae (Scarabaeinae) are

represented by very few specimens (1 to 4), while in the study of Frank et al. (2017) (done in the same area), these same species were collected with a notable greater abundance, which seems to indicate that they are coprophagous (aspect that I have also read in several works) and their collection in carrion is occasional. Unless the authors provide convincing information that these species are copro-necrophagous or important reasons not to exclude them from the study. The seven collected species of Aphodiidae and Scarabaeidae (Scarabaeinae) represent only 0.14% of the 10,488 specimens collected and the 99.86% correspond to the two geotrupid species. Species collected on carrion of Scarabaeidae and Aphodiidae are included and analyzed by Frank et al. (2017) in their study of dung beetles associated to dung as part of the same big German project. Also, the two geotrupid species are included in that study, but their abundances were similar to the abundances reported in the present manuscript, which is not discussed, but I think is an evidence that these species are in fact copro-necrophagous, and their analyses, as part of dung beetles associated to dung and carrion, is important, because they are two of the most abundant species with a strong impact in the dung and carrion ecology. If the study focuses only on Geotrupidae, the confusion I noticed throughout the text will be avoided, where Scarabaeoidea is referred to as equivalent to dung beetles; however, Scarabaeoidea includes a large amount of taxa, many of which are not dung beetles.

3. Introduction, page 2, lines 52-53: "During the 52 so-called 'post-bloating stage' of cadaver decomposition, the large feeding masses of fly maggots are joined by 53 predatory coleopterans, i.e. by members of the Silphidae, Staphylinidae and Histeridae." The majority of the silphid species are necrophagous, I think the word "some members of" can help (not all staphylinid species are predatory, but yes the majority).

Page 2, line 62: If it is pertinent to the journal, the author and year of each species could be added only the first occasion that is written.

Page 3, lines 88-89: "Our results can be generalized, as our study encompasses three regions differing in their geology, topography and climate." This statement could be debatable, because the biogeographic history is not the same for any place in the World. Maybe "Our result can be generalized in Germany" or "in Central Europe".

4. Methods and Materials, page 4, line 127: "...important factors that affect the abundance of copro-necrophagous dung beetles." As commented previously, the Aphodiidae and Scarabaeidae species collected in this work probably do not are copro-necrophagous.

5. Results, page 6, lines 208 and 216: "Influence of environmental variables on the abundance of the single scarabaeoid species *A. stercorosus*", and "Influence of environmental variables on the abundance of the single scarabaeoid species *T. vernalis* in the Schorfheide-Chorin region", change "scarabaeoid" for "geotrupid".

6. Discussion, page 7, lines 271 and 272: "From all other taxa of Scarabaeidae and Aphodiidae, we registered only 15 specimens altogether, compared with 10473 specimens in the Geotrupidae taxon." Or it may be that these particular species are not carrion eaters but of dung. Here you probably need compare the abundance of this species with the results of Frank et al. (2017). The great abundance of the two geotrupid species also is reported in that paper (coinciding with the copro-necrophagous habits).

Page 7, line 278: "Hence, the large number of coprophagous dung beetles might substantially contribute to carrion decomposition...". Coprophagous or copro-necrophagous? I have the impression that there is no distinction between one eating habit and another.

Page 7, lines 284-285: "We reported similar variations in our previous study on silphid beetles visiting cadavers within the Biodiversity Exploratories." The authors have previously studied the silphids, and in this contribution Geotrupidae (and probably some Aphodiidae and scarabs) associated to carrion; other colleagues have studied the Aphodiidae, Scarabaeidae and Geotrupidae using dung. I have two observation in relation with this magnificent knowledge: do not consider the fact of competition between these groups affecting the results (at least their consideration to future studies) and do not integrate completely (or compare) the results of

Geotrupidae, Aphodiidae and Scarabaeidae obtained using dung with respect to results using carrion.

Page 8, lines 287-289: "A comprehensive survey of scarabaeoid beetles attracted to dung found that, overall, dung beetle biomass was 80 times higher in the Schorfheide than in the two other regions [23]." I am not sure if Frank et al. (2017) say the same that you say. I cannot access to the paper (would have to buy it), but yes at the Ph. D. thesis of Frank, and I am not sure that the text are the same in the thesis and the paper, but I think yes. In the thesis is written "Overall, dung beetle biomass was 10 times higher in forests than in grasslands in the Alb (Welch t-test, $t = 3.83$, $p < 0.001$), 20 times in the Hainich ($t = 5.62$, $p < 0.001$), up to 80 times in the Schorfheide ($t = 8.99$, $p < 0.001$) (Fig. 2b)." (Chapter 2, page 20). The comparison is between forest and grasslands in each site, not between sites.

Page 9, lines 359-360: "As *A. stercorosus* and *T. vernalis* are tunnellers and primarily bury dung [78], ...". This reference (Anduaga, 2004) do not speak about these species, because focused on dung beetles from a place of Durango, Mexico, and this geotrupid species do not exist in Mexico. Page 10, lines 396-397: "We have suggested that this reduction of the Simpson's dominance in predacious silphid beetles alters the calculable...". All studied silphid species are predaceous? I think not, I think the majority are carrion eating.

7. Conclusions, page 11, lines 411-412: "Thus, scarabaeoid beetles caught on animal carrion can be no longer considered mere by-catch." This is true for the two geotrupid species, but I think not for any Scarabaeoidea species; therefore, not for any Scarabaeidae and Aphodiidae species.

Page 11, lines 416-417: "For instance, do adults of certain dung beetle species depend on cadavers as a food source or is their interest in cadavers plain opportunism?" What response can suggest the fact that the abundance of the two geotrupid species in dung (Frank et al. 2017) and carrion are near similar? Maybe it is a good place to do a question about the effect of competence between Silphidae, Geotrupidae, Scarabaeidae and Aphodiidae for the dung and for the carrion in their abundance, richness and diversity. Another question: how do found results vary throughout the seasons?

Author's Response to Decision Letter for (RSOS-191722.R0)

See Appendix A.

RSOS-191722.R1 (Revision)

Review form: Reviewer 2

Is the manuscript scientifically sound in its present form?

Yes

Are the interpretations and conclusions justified by the results?

Yes

Is the language acceptable?

Yes

Do you have any ethical concerns with this paper?

No

Have you any concerns about statistical analyses in this paper?

No

Recommendation?

Accept as is

Comments to the Author(s)

Thank you for your detailed and polite responses. I am very satisfied with the manuscript, and I congratulate you on your excellent research.

Review form: Reviewer 3 (Juan Márquez)

Is the manuscript scientifically sound in its present form?

Yes

Are the interpretations and conclusions justified by the results?

Yes

Is the language acceptable?

Yes

Do you have any ethical concerns with this paper?

No

Have you any concerns about statistical analyses in this paper?

No

Recommendation?

Accept as is

Comments to the Author(s)

I reviewed all point by point responses of the authors, particularly responses to my comments, but also all responses, and I agree with the correction made by the authors in the new version of the paper.

Decision letter (RSOS-191722.R1)

03-Feb-2020

Dear Dr von Hoermann,

It is a pleasure to accept your manuscript entitled "Forest habitat parameters influence abundance and diversity of cadaver-visiting dung beetles in Central Europe" in its current form for publication in Royal Society Open Science. The comments of the reviewer(s) who reviewed your manuscript are included at the foot of this letter.

Please ensure that you send to the editorial office an editable version of your accepted manuscript, and individual files for each figure and table included in your manuscript. You can send these in a zip folder if more convenient. Failure to provide these files may delay the

processing of your proof. You may disregard this request if you have already provided these files to the editorial office.

on behalf of Kevin Padian (Subject Editor)
openscience@royalsociety.org

Associate Editor Comments to Author:
Congratulations on the acceptance of your work.

Reviewer comments to Author:
Reviewer: 3

Comments to the Author(s)
I reviewed all point by point responses of the authors, particularly responses to my comments, but also all responses, and I agree with the correction made by the authors in the new version of the paper.

Reviewer: 2

Comments to the Author(s)
Thank you for your detailed and polite responses. I am very satisfied with the manuscript, and I congratulate you on your excellent research.

Appendix A

Dear Andrew Dunn and Prof. Kevin Padian,

Thank you for the reviewer's comments on our previous version of the manuscript ID RSOS-191722. The critique and suggestions have been very helpful and greatly improved our manuscript. In this revised manuscript, we were able to address the reviewer's comments. We improved the reporting of the aim/hypotheses, methodological aspects and biotic/abiotic parameters included in our analyses. We furthermore increased the focus on our two main species (representing the Geotrupidae) as suggested by reviewer 3. Please find below a detailed point-by-point response to the comments of the referees. In the revised manuscript (von Hoermann et al. RSOS_tracked_changes.doc) changes in the text are highlighted in purple (reviewer #1), blue (reviewer #2) and green (reviewer #3).

For all the authors I am yours sincerely,

Christian v. Hoermann

>>>>>>>>>>>>>>

Comments to Author:

Reviewers' Comments to Author:

Reviewer: 1

Comments to the Author(s)

Dear authors;

The manuscript focus on a very interesting topic, which is the influence of abiotic and biotic factors in the diversity of copronecrophagous beetles. Also, the study focus on several important topics associated to ecosystem process such as the carrion decomposition, which is unusual because most of studies with Scarabaeidae beetles evaluated the dung decomposition. Data and analysis performed by authors seems to be the appropriate for this kind of studies. After I read the manuscript, there are some major issues that should be addressed before this manuscript can be published:

Major comments

1. Authors should include the aim and hypothesis of your work.

We agree with the reviewer and followed his advice by providing more detailed information on the aim and hypothesis of our work, see end of the introduction section, line 108-116.

2. I consider that table 1, table 2, table 3 and figure 1 could be in electronic appendix. Besides, I suggest that figure 3b and 3c, and figure 5d do not be published because these values are nor statistical significance ($p>0.05$).

We followed the reviewer's advice and included table 1-3 in the electronic appendix. We also removed figure 3b and c, as well as figure 5d.

3. I think could be interesting if authors include more information about the ecology/biology of these two main species they found, which seems the dominant species in central region of Europe.

We agree with the reviewer and included a comprehensive description of the biology and ecology of the two most abundant species *A. stercorosus* and *T. vernalis* in the introduction section (lines 70 – 85).

4. Be careful with the use of not statistical significance values ($p > 0.05$); I understand that is very common the use of that information as “tendency” o “patterns”.

We understand the reviewer’s considerations and stated “tendencies” more clearly; see major comment 2 and point-by-point answers in the minor comments on the result section.

Recommended literature. We included the following helpful references.

Dortel, E., Thuiller, W., Lobo, J. M., Bohbot, H., Lumaret, J. P. & Jay-Robert, P. (2013). Potential effects of climate change on the distribution of Scarabaeidae dung beetles in Western Europe. *Journal of Insect Conservation*, 17, 1059-1070.

<https://doi.org/10.1007/s10841-013-9590-8>

Giménez, V. C., Verdú, J. R., Guerra Alonso, C. B. & Zurita, G. A. (2018). Relationship between land uses and diversity of dung beetles (Coleoptera: Scarabaeinae) in the southern Atlantic forest of Argentina: which are the key factors? *Biodiversity and Conservation*, 27, 3201-3213. <https://doi.org/10.1007/s10531-018-1597-8>

Neita, J. C. & Escobar, F. (2012). The potential value of agroforestry to dung beetle diversity in the wet tropical forests of the Pacific lowlands of Colombia. *Agroforestry Systems*, 85, 121-131. <https://doi.org/10.1007/s10457-011-9445-9>

Verdú, J. R. & Lobo, J. (2008). Ecophysiology of thermoregulation in endothermic dung beetles: Ecological and geographical implications. *Insect Ecology and Conservation*, 661, 1-28. ISBN: 978-81-308-0297-8

Minor comments

Abstract

Lines 30-31: I think this is wrong. The highest beetle diversity was associated to Shannon total vascular plants (both, Shannon and Simpson) and fine sands (only Simpson); whereas total abundance was associated to mean ambient temperature (see Table 1 and 3). However, the last result is very important because copronecrophagous beetles are highly sensitive to changes in temperature and humidity, due to their physiological restrictions (Neita and Escobar, 2012; Verdú and Lobo, 2008; Dortel et al., 2013; Giménez Gómez et al., 2018).

We changed the line to include the respective information as follows: “*High beetle abundance was associated with higher mean ambient temperature. Furthermore, A. stercorosus and T. vernalis were more abundant in areas where soil contained higher portions of fine sand.*”

Introduction

Lines 83-86: this is the aim of your study? This seems like a short description of the activities you made. I suggest that rephrased this statement, try to be more specific about what was the aim of this study and included the hypothesis

As stated in major comment 1, we agree with the reviewer and provide more detailed information on the aim and hypothesis of our work, see end of the introduction section, line 108-116.

Lines 86-87: this should be in the results section; Lines 87-88: this should be in the methods section; Lines 88-89: this should be in the discussion/conclusions section

We agree with the reviewer that these sentences should not be included in the Introduction. We entirely removed the respective lines and replaced the section with more detailed hypotheses (as stated in the comment above).

Methods

Lines 101-102: Your study area were the Biodiversity Exploratories, but, you compared three regions; so, I suggest that changed “study area” to “region” from here to the end of this section

We made the requested alterations in the text.

Results

Lines 178-179: I suggest “we captured 10488 individuals from the following nine scarabaeoid beetles species”; or you can use a similar statement of lines 182-183

We changed the statement in lines 227 – 228 according to the reviewer’s suggestion.

Line 189: I suggest that change “importance” to “effects”

We followed the suggestion of the reviewer (see line 238).

Lines 196-197, 217-218, 229-230, 240-241: Avoid “introduction statements” and just go directly with your results

The reviewer is right. The introduction statements gave no additional information values. We deleted all “introduction statements” in lines 244 – 245, 265 – 266, 280 – 281 and 291 – 292.

Lines 199-200: I do not agree with authors. I suggest that analyzed your results if it has statistical differences ($p < 0.05$). Besides, the influence of mean ambient temperature is a great results because copronecrophagous beetles are highly sensitive to changes in temperature and humidity, due to their physiological restrictions (Neita and Escobar, 2012; Verdú and Lobo, 2008; Dortel et al., 2013; Giménez Gómez et al., 2018)

We followed the suggestion of the reviewer and removed the results analysis of the not significant variables ‘soil moisture’ and ‘fine sand’ (see line 248 – 252). We also included the very helpful references regarding physiological restrictions at the end of the introduction section in lines 114 – 116.

Lines 200-201: I suggest that unify this statement in lines 198-199

We unified the statement in lines 246 – 248.

Lines 201-202: Again, I do not agree with authors. Soil moisture had not effect on overall beetle abundance ($p = 0.08$)

We agree with the reviewer and removed the respective sentence in lines 251 – 253.

Lines 210-212: I found this statement redundant. Authors should simplify this

The reviewer is right. We simplified the corresponding statement in lines 257 – 260.

Lines 220-221: I do not agree with authors ($p > 0.05$)

We removed the statements regarding the not significant variable ‘soil moisture’ in lines 273 – 274.

Lines 217-224: This is very redundant and complex. Authors should simplify this as “The total abundance of *T. vernalis* increased with higher fine-sand contents, higher vascular plant diversity (Shannon and Simpson’s diversity) and for stand density of the main tree species

(table 1c); in contrast, the abundance of *T. vernalis* decreased with higher forest stand-age (figure 4b, table 1c) (quasi-Poisson-GLM: $F=53.97$, $p<0.001$).”

We agree with the reviewer and simplified the whole section regarding the recommended formulation (see lines 267 – 278).

Line 225: I do not agree with authors ($p>0.05$)

We removed the sentence in line 278 because of not significant values for ‘soil moisture’.

Lines 232-233: I do not agree with authors ($p=0.06$)

We removed the sentences in lines 285 – 286 and 288 – 289 because of not significant values for ‘fine sand’.

Lines 229-234: I think this is a redundant statement

The reviewer is right. We unified the section by deleting the sentences in lines 282 – 286.

Line 235: I do not agree with authors ($p=0.06$)

We removed the sentence in lines 288 – 289 because of a not significant value for ‘fine sand’.

Lines 243-244: I do not agree with authors ($p=0.07$)

We removed the sentence in lines 297 – 298 because of a not significant value for ‘dbh standard deviation’.

Lines 245-246: I do not agree with authors ($p=0.07$)

We removed the corresponding sentence in lines 300 – 301.

Lines 240-246: Authors should rephrased this statement because it is redundant

We rephrased the statement by deleting lines 294 – 298.

Lines 247:250: I do not agree with authors ($p=0.08$)

We agree with the reviewer and changed the statement ‘a tendential difference’ in ‘no significant difference’ in line 302.

Lines 252-253: Authors should unify this lines

We unified the lines by deleting lines 304 – 306.

Discussion

Line 268: ScarAbaeoid

We thank the reviewer for the linguistic hint and changed the word in ‘Dung’ as requested by reviewer #3.

Reviewer: 2

Comments to the Author(s)

Review of RSOS-191722 “Forest habitat parameters influence abundance and diversity of cadaver-visiting dung beetles”.

LL83-89. The questions being asked are not clear. Although the broad aim is given, I would like to see a list of questions that are answered in this study.

We agree with the reviewer and (following reviewer 1) we provide more detailed information on the aim and hypothesis of our work, see end of the introduction section, line 108-116.

L116. please provide some details on the pitfall trapping method. how long were they opened for? what preservative did you use? what diameter? these details should be in this paper.

We included the requested details on the pitfall trapping method in the ‘Methods and Materials’ section in lines 156 – 158.

L126. Some further detail is needed here, without reference to the supporting material. For example, were variables collected at each carcass? or were they averaged across each 100 x 100 plot? Why were so many variables examined? There seems no clear link between the variables and expected dung beetle abundances/occurrences.

We agree with the reviewer and included some further details regarding variable collection (variables were averaged across each 100 x 100 m plot based on multiple inventory campaigns performed within the Biodiversity Exploratories) in lines 176 – 179. Additionally, based on physiological restrictions and living habits of dung beetles, we established a clear link between all the examined variables and expected dung beetle abundance/occurrences (as requested by reviewer #1 as well) in lines 108 – 116 at the end of the introduction section.

L140. GLMs are univariate approaches (you are investigating predictors of individual response variables).

The reviewer is right. Because we are investigating predictors of individual response variables (like abundance or Shannon’s diversity), we changed the term ‘multivariate’ in ‘univariate’ in line 183 of the ‘statistics’ section.

L151. what trap types? I can only see pitfall traps were used, please clarify.

In lines 150 – 151 we included the information that we used controls (pitfall traps without cadavers and wire cages) as an additional trap type. In lines 200 – 202, we included the additional information that piglet cadaver traps were compared with control traps.

L164. What is the question being addressed here? This approach seems very much a fishing exercise, with all sorts of variables examined to find a potential relationship. I think that a revision of the methods would be very useful to explain your choice of variables, and how this relates to your questions.

As requested by reviewer #1 as well, we explained the choice of variables along with our research questions at the end of the introduction section in lines 110 – 116.

L169. this paragraph seems added at the last minute and doesn't quite fit here. these response variables would be better described earlier at the beginning of the stats section.

The reviewer is right. We shifted the description of the response variables to the beginning of the stats section (see lines 186 – 190) and removed the last paragraph beginning in line 218.

L189. here (and elsewhere) consider replacing 'degradation phase' with 'decay stage', as this is more commonly used and understood.

We changed the terminology 'degradation phase' in 'decay stage' in line 238.

L200. what does 'tendentally guilty' mean? perhaps a translation issue.

As requested by reviewer #1, we deleted all statements describing or interpreting tendencies throughout the manuscript.

General comment: I wonder how mean temperature changed with decay stage? If days were warmer, this might also explain higher abundance during later decay.

We agree with the reviewer that warmer days cause higher emission rates of cadaveric volatile organic compounds and consequently higher numbers of carrion associated insect visitors should be attracted. However, the following exemplary temperature profiles of our data loggers (directly mounted inside of the wire cages containing our exposed piglets) show very constant ambient temperature sequences during August 2014 in all three exploratories (Alb, Hainich and Schorfheide) in Germany:

piglet #34 in Hainich region

Piglet #47 in Hainich region

Piglet #15 in Alb region

Piglet #22 in Alb region

Piglet #53 in Schorfheide region

Piglet #66 in Schorfheide region

LL196-256. Results describing environmental effects on beetle diversity and abundance of individual species could be reduced in length. It is currently a bit repetitive, and the authors might consider looking for ways to shorten this section to highlight only the key results.

We agree with the reviewer and extensively shortened and simplified the results describing environmental effects on beetle diversity and abundance as requested by reviewer #1 as well (see lines 244 – 311).

L259. unclear what this ‘framework’ is. Do you mean your study design?

With this ‘framework’, we mean our study design of the superordinate ‘NecroPig’-project that is described in more detail in von Hoermann et al. (2018). For avoiding confusion, we modified the sentence in: “Data from dung beetles captured on 61 decomposing piglets in variously managed forests were compiled in three regions of the German republic” in lines 314 – 315.

L332. I am still unsure what the land uses are. In your methods you say that all sites were subject to forestry management, so what are the measures of land use and intensity? These variables are distinct from local habitat/environmental variables you examined, and so should be a separate question, yes?

We thank the reviewer for the hint to give additional information what the measures of land use and intensity are and explained them in much more detail in the methods section (subsection ‘Study sites and piglet cadaver exposition’) in lines 133 – 143.

LL411-435. I think that the conclusions do not summarise the key findings very well. What is new and different about his study that no one else has described? It is not surprising that you found lots of scarabs at pig carcasses. However, I think it is interesting they had highest abundance in later decay, and that some species were more abundant than others. Clearly describing the main results will help to the reader to see how your study has added to the literature.

We agree with the reviewer and added more explicitly our new findings that are different to other study outcomes, like the highest dung beetle abundance in later decay and the huge difference in abundance between the taxa Geotrupidae, Aphodiidae and Scarabaeidae (lines 480 – 487).

Reviewer: 3

Comments to the Author(s)

1. Title: I suggest being specific: "in Germany", because the results could not be the same in other parts of the World.

We changed the title in: 'Forest habitat parameters influence abundance and diversity of cadaver-visiting dung beetles in Central Europe'. Our modeling results are not extremely specific for Germany but rather for Central Europe, where similar habitat conditions can be adopted.

2. Throughout the entire document: I think the paper can focus only on the two species of Geotrupidae, because the species collected from Aphodiidae and Scarabaeidae (Scarabaeinae) are represented by very few specimens (1 to 4), while in the study of Frank et al. (2017) (done in the same area), these same species were collected with a notable greater abundance, which seems to indicate that they are coprophagous (aspect that I have also read in several works) and their collection in carrion is occasional. Unless the authors provide convincing information that these species are copro-necrophagous or important reasons not to exclude them from the study.

The seven collected species of Aphodiidae and Scarabaeidae (Scarabaeinae) represent only 0.14% of the 10,488 specimens collected and the 99.86% correspond to the two geotrupid species.

Species collected on carrion of Scarabaeidae and Aphodiidae are included and analyzed by Frank et al. (2017) in their study of dung beetles associated to dung as part of the same big German project. Also, the two geotrupid species are included in that study, but their abundances were similar to the abundances reported in the present manuscript, which is not discussed, but I think is an evidence that these species are in fact copro-necrophagous, and their analyses, as part of dung beetles associated to dung and carrion, is important, because they are two of the most abundant species with a strong impact in the dung and carrion ecology.

If the study focuses only on Geotrupidae, the confusion I noticed throughout the text will be avoided, where Scarabaeoidea is referred to as equivalent to dung beetles; however, Scarabaeoidea includes a large amount of taxa, many of which are not dung beetles.

We agree with the reviewer and explicitly mentioned in the 'Statistics' section in Methods and Materials the huge percentage of individuals of the geotrupid species *A. stercorosus* and *T. vernalis* at our exposed piglet cadavers compared to the very few individuals out of the Aphodiidae and Scarabaeidae (lines 191 – 193). Therefore, we further defined the application of separate statistical analyses for these two most abundant taxa. Additionally, in the discussion section (lines 329 – 334), we included the very similar study results of Frank et al. (2017) for coprophagous dung beetles in the Biodiversity Exploratories. In this context we picked up the debate of coprophagy and true copro-necrophagy as suggested by the reviewer.

We mentioned in lines 339 – 342, that our results-discussion will focus on the two very abundant copro-necrophagous species of Geotrupidae. We agree with the reviewer that copro-necrophagy is only "secured" in the Geotrupidae taxon based on our findings and on the findings of Frank et al. (2017). Throughout the manuscript, in the context of *A. stercorosus* and *T. vernalis*, we substituted the too broadly described term 'scarabaeoid beetles' by 'geotrupid beetles'. We agree with the reviewer that the taxon Scarabaeoidea comprises a lot of beetles that neither feed on dung nor on carrion and mentioned this aspect in lines 339 – 342 in the discussion section. In general, throughout the manuscript, we substituted the term 'scarabaeoid' by 'dung beetles' or rather 'scarabaeoid dung beetles'.

3. Introduction, page 2, lines 52-53: "During the 52 so-called 'post-bloating stage' of cadaver decomposition, the large feeding masses of fly maggots are joined by 53 predatory

coleopterans, i.e. by members of the Silphidae, Staphylinidae and Histeridae.” The majority of the silphid species are necrophagous, I think the word "some members of" can help (not all staphylinid species are predatory, but yes the majority).

We agree with the reviewer and changed 'by members of' to 'some members of' in line 55.

Page 2, line 62: If it is pertinent to the journal, the author and year of each species could be added only the first occasion that is written.

We added author and year for both geotrupid species in line 65.

Page 3, lines 88-89: “Our results can be generalized, as our study encompasses three regions differing in their geology, topography and climate.” This statement could be debatable, because the biogeographic history is not the same for any place in the World. Maybe "Our result can be generalized in Germany" or "in Central Europe".

We entirely removed the respective statement and completed the section with more detailed hypotheses (as stated by reviewer #1).

4. Methods and Materials, page 4, line 127: "...important factors that affect the abundance of copro-necrophagous dung beetles." As commented previously, the Aphodiidae and Scarabaeidae species collected in this work probably do not are copro-necrophagous.

We agree with the reviewer and changed the sentence in: '... important factors that affect the abundance of dung beetles attracted to carrion (lines 167 – 168).

5. Results, page 6, lines 208 and 216: “Influence of environmental variables on the abundance of the single scarabaeoid species *A. stercorosus*”, and “Influence of environmental variables on the abundance of the single scarabaeoid species *T. vernalis* in the Schorfheide-Chorin region”, change “scarabaeoid” for “geotrupid”.

We changed 'scarabaeoid' for 'geotrupid' in lines 257 and 265.

6. Discussion, page 7, lines 271 and 272: “From all other taxa of Scarabaeidae and Aphodiidae, we registered only 15 specimens altogether, compared with 10473 specimens in the Geotrupidae taxon.” Or it may be that these particular species are not carrion eaters but of dung. Here you probably need compare the abundance of this species with the results of Frank et al. (2017). The great abundance of the two geotrupid species also is reported in that paper (coinciding with the copro-necrophagous habits).

We compared the abundance numbers with the results of Frank et al. (2017). Regarding the coincidence with the copro-necrophagous habit, we added the following sentence: line 332-334 “Consequently, based on such high trapping numbers at carrion and dung resources as well, we designate both geotrupid species as true copro-necrophagous, having a strong impact on dung and carrion ecology.” See our detailed response to the statement No. 1 of the reviewer, fitting to this statement No. 6.

Page 7, line 278: “Hence, the large number of coprophagous dung beetles might substantially contribute to carrion decomposition...”. Coprophagous or copro-necrophagous? I have the impression that there is no distinction between one eating habit and another.

This was a linguistic error from our point of view. We changed it in: “Hence, the large number of copro-necrophagous geotrupid beetles might substantially contribute to carrion decomposition ...” (line 337).

Page 7, lines 284-285: "We reported similar variations in our previous study on silphid beetles visiting cadavers within the Biodiversity Exploratories." The authors have previously studied the silphids, and in this contribution Geotrupidae (and probably some Aphodiidae and scarabs) associated to carrion; other colleagues have studied the Aphodiidae, Scarabaeidae and Geotrupidae using dung.

I have two observations in relation with this magnificent knowledge: do not consider the fact of competition between these groups affecting the results (at least their consideration to future studies) and do not integrate completely (or compare) the results of Geotrupidae, Aphodiidae and Scarabaeidae obtained using dung with respect to results using carrion.

We added the fact of competition as well as dung and carrion beetle results integration as a future perspective for the modelling in carrion ecology studies in lines 367 – 373.

Page 8, lines 287-289: "A comprehensive survey of scarabaeoid beetles attracted to dung found that, overall, dung beetle biomass was 80 times higher in the Schorfheide than in the two other regions [23]." I am not sure if Frank et al. (2017) say the same that you say. I cannot access to the paper (would have to buy it), but yes at the Ph. D. thesis of Frank, and I am not sure that the text are the same in the thesis and the paper, but I think yes. In the thesis is written "Overall, dung beetle biomass was 10 times higher in forests than in grasslands in the Alb (Welch t-test, $t = 3.83$, $p < 0.001$), 20 times in the Hainich ($t = 5.62$, $p < 0.001$), up to 80 times in the Schorfheide ($t = 8.99$, $p < 0.001$) (Fig. 2b)." (Chapter 2, page 20). The comparison is between forest and grasslands in each site, not between sites.

We thank the reviewer for the important hint and corrected and expanded our statement based on the findings from Frank et al. (2017) by mentioning the multiplying factors of beetle biomass between forests and grasslands, separately for each exploratory, in lines 351 – 353.

Page 9, lines 359-360: "As *A. stercorosus* and *T. vernalis* are tunnellers and primarily bury dung [78], ...". This reference (Anduaga, 2004) do not speak about these species, because focused on dung beetles from a place of Durango, Mexico, and this geotrupid species do not exist in Mexico.

We thank the reviewer for the important hint and replaced the reference to Anduaga (2004) by a new reference to Frank et al. (2017) in lines 428 – 430.

Page 10, lines 396-397: "We have suggested that this reduction of the Simpson's dominance in predacious silphid beetles alters the calculable...". All studied silphid species are predaceous? I think not, I think the majority are carrion eating.

We modified the sentence to: "We have suggested that ... in predacious and/or carrion eating silphid beetles alters the calculable ..." in line 465.

7. Conclusions, page 11, lines 411-412: "Thus, scarabaeoid beetles caught on animal carrion can be no longer considered mere by-catch." This is true for the two geotrupid species, but I think not for any Scarabaeoidea species; therefore, not for any Scarabaeidae and Aphodiidae species.

We took into account the argument of the reviewer and modified the sentence to: "Thus, dung beetles, in particular the most abundant geotrupid species *A. stercorosus* and *T. vernalis*, caught on animal carrion ..." in lines 480 – 481.

Page 11, lines 416-417: "For instance, do adults of certain dung beetle species depend on cadavers as a food source or is their interest in cadavers plain opportunism?" What response can suggest the fact that the abundance of the two geotrupid species in dung (Frank et al. 2017) and carrion are near similar? Maybe it is a good place to do a question about the effect

of competence between Silphidae, Geotrupidae, Scarabaeidae and Aphodiidae for the dung and for the carrion in their abundance, richness and diversity. Another question: how do found results vary throughout the seasons?

We added the additional questions suggested by the reviewer in lines 492 – 496 of the 'Conclusion' section.